# Northern Hemisphere in situ snow water equivalent dataset (NorSWE, 1979-2021)

Colleen Mortimer[1], Vincent. Vionnet[2]

[1]Climate Research Division, Environment and Climate Change Canada, Toronto, Canada
[2]Meteorological Research Division, Environment and Climate Change Canada, Dorval, Canada

*Correspondence to*: Colleen Mortimer (colleen.mortimer@ec.gc.ca)

**Abstract.** In situ observations of snow water equivalent (SWE) are critical for climate applications and resource management yet there is no global database of in situ SWE observations. Here, we present the Northern Hemisphere in situ snow water equivalent dataset (NorSWE) consisting of over 11.5 million SWE observations from more than 10 thousand different locations
across the Northern Hemisphere spanning the modern satellite era (1979–2021). NorSWE builds on an existing framework applied to Canadian data (CanSWE; Vionnet et al., 2021). It includes SWE observations from manual snow courses covering Canada, the United States, Norway, Finland and Russia and from automated sensors (snow pillows, snow scales, and automated passive gamma radiation sensors) in Canada, the United States, Norway, and Nepal. Airborne passive gamma SWE estimates provide additional coverage over North America. Exceptionally, to expand coverage over Europe we also include single point
manual SWE observations from eleven sites in Switzerland. In addition to SWE, snow depth (SD) and derived bulk snow density are included when available. A consistent quality control is applied to all records and the final dataset delivered as a single NetCDF file that is publicly available at https://doi.org/10.5281/zenodo.15263370 (Mortimer and Vionnet, 2025).

## 1 Introduction

Accurate knowledge of the amount of water stored in the seasonal snowpack is critical for risk and resource management
including flood and drought forecasting (e.g. Barnett et al., 2005; Fyfe et al., 2017; Huning et al., 2020; Vionnet et al., 2020), water supply for agriculture (Biemans et al., 2019; Qin et al., 2020) and human consumption (Foster et al., 2011, Sturm et al., 2017), hydropower operations (Magnusson et al., 2020), as well as for ecosystems and climate monitoring (Meredith et al., 2019; Thornton et al., 2021; Gottlieb and Mankin, 2024). It is quantified by the snow water equivalent (SWE), or water equivalent of the snow cover, which is *'the vertical depth of water that would be obtained if the snow cover melted completely,*
*which equates to the snow-cover mass per unit area'* (WMO, 2018). At global scales, SWE can be estimated from climate models (Mudryk et al., 2020), reanalyses (Mudryk et al., 2025 and references therein) and, to some extent, from passive microwave satellite observations (Pulliainen et al., 2020) but these methods usually produce SWE estimates at medium to coarse spatial resolutions (5–25km) and their accuracy must be verified against ground-based measurements. In situ, SWE can be measured manually from snow pits or using a snow tube (WMO, 2018), from automated sensors such as snow pillows

(Beaumont, 1965), snow scales (Johnson, 2004; Smith et al., 2017), passive gamma radiation sensors (Kodama et al., 1979; Paquet et al., 2008), or the analysis of GNSS signals (Henkel et al., 2018; Steiner et al., 2022). SWE can be estimated from in situ snow depth (SD, formally abbreviated as HS (WMO, 2018)) measurements using snow density models relating SD to SWE, as in the Northern Hemispheric dataset NH-SWE (Fontrodona-Bach et al., 2023) and the airborne lidar-based Airborne Snow Observatory (ASO, Painter et al., 2016); however, these are not direct observations of SWE. Rather, they rely on

ancillary data and direct in situ observations, such as those provided in our dataset, for their development.

Snow cover varies spatially, influenced by landcover and microclimates, so SWE measured at a single point or from automated sensors, which have footprints $\sim< 100$ m$^2$, may not be representative of a larger area (López-Moreno et al., 2020; Meromy et al., 2013). Therefore, for manual SWE observation, it is common practice to collect multiple SWE and SD measurements along a predefined route, referred to as a multi-point gravimetric snow survey, a snow course, or a snow transect (WMO,

2018). These multiple measurements are averaged together to provide a single SWE value for the area of interest. At larger scales still, SWE is estimated from airborne surveys of passive gamma radiation by relating the attenuation of gamma radiation emitted from the upper layers of the soil by the intervening water mass (solid or liquid) after accounting for the background soil moisture (Carrol, 2001). This principle, also employed by automated gamma radiation sensors (e.g. Choquette et al., 2013), has been used operationally in the United States (NOHRSC) since 1979. Direct comparisons of airborne gamma and snow

course measurements showed reasonable correspondence (correlation > 0.7) in non-mountain areas up to distances of at least 50 km (Mortimer et al., 2024).

Although manual in situ SWE measurements have been conducted for nearly a century (USDA, 2008; Bulygina et al., 2011) there is no standard global database to archive these observations, nor is there a standard approach to measuring SWE (Pirazzini et al., 2018 for Europe). Some national agencies such as the All-Russia Research Institute of Hydrometeorological Information

– World Data Center (RIHMI-WDC) and the Finish Environmental Institute (SYKE) maintain a comprehensive national network of repeated manual snow surveys whose data are archived and searchable. Elsewhere, such as Canada and the United States, SWE is measured separately by various agencies, government departments, and hydropower companies, some of which are consolidated into larger databases for example the Canadian historical Snow Water Equivalent dataset (CanSWE) (Vionnet et al., 2021) and by the Natural Resources Conservation Service (NRCS) through its regional data collection offices (Fleming

et al., 2023). It is under this fragmented landscape that we compiled available in situ SWE measurements spanning North America, Norway, Finland, Switzerland, Russia, and Nepal into a single dataset the Northern Hemisphere in situ snow water equivalent dataset (NorSWE, Mortimer and Vionnet, 2025). NorSWE was initially compiled to support the evaluation of gridded SWE products (Elias Chereque et al., 2024; Mortimer et al., 2024; Mudryk et al., 2025) so we concentrated on compiling snow courses and airborne gamma SWE which are more spatially representative than automated instruments. To

support evaluation of hydrological models (e.g. Arnal et al., 2024) which requires a higher temporal frequency than available from the snow courses, we added automated SWE measurements from snow pillows, snow scales and automated passive gamma radiation sensors over North America, Norway, and Nepal. To fill data gaps in Europe we exceptionally include single point manual SWE and SD observations from the Swiss GCOS network (Marty, 2020). NorSWE covers the period 1979 –

2021 and is available as a single NetCDF file following the conventions of CanSWE (Vionnet et al., 2021). Dataset
development generally followed Vionnet et al. (2021), with additional procedures and data attributes to support the non-Canadian data sources described herein. This paper is organized as follows: Section 2 describes the types of measurements included in the dataset, Sections 3 through 5 outline the data processing and quality control (QC) procedures, Sections 6 summarizes the published dataset, Sections 7 and 8 discuss its usage and limitations, and a brief conclusion is given in Section 9.

## 2 SWE measurement methods included

NorSWE includes SWE observations from manual gravimetric snow surveys, airborne gamma SWE, automated snow pillows, snow scales and passive gamma radiation sensors from the sources listed in **Table 1**. Exceptionally, it also includes single point manual observations from select sites in Switzerland. The measurement type codes (**Table 2**) follow the WMO BUFR table (WMO, 2019) except for airborne gamma which we assign code 64 to differentiate it from passive gamma radiation sensors (**Table 3**).

**Table 1: Data sources included in NorSWE.**

| Data provider or dataset | Station ID prefix | Geographic domain | Measurement method(s) | Variables | Data access |
|---|---|---|---|---|---|
| CanSWEv6 [Environment and Climate Change Canada and partners] | CanSWE | Canada | Snow course, snow pillows and snow scales, passive gamma radiation sensors, acoustic SD sensors* | SWE, SD. Bulk density derived from SWE and SD | https://doi.org/10.5281/zenodo.10835278 accessed February 2024. |
| U. S. Department of Agriculture Natural Resources Conservation Service (NRCS) – snow survey | NRCS | Western US and Alaska | Snow course. Aerial markers excluded. | SWE, SD. Bulk density derived from SWE and SD | https://www.nrcs.usda.gov/wps/portal/wcc/home/snowClimateMonitoring/snowpack/ accessed February 2022. |
| U. S. Department of Agriculture Natural Resources Conservation – SNOTEL | SNOTEL | Western US and Alaska | Snow pillows and snow scales, acoustic SD sensors*. | SWE, SD. Bulk density derived from SWE and SD | Direct download using soilDB (Beaudette et al., 2024) https://CRAN.R-project.org/package=soilDB accessed April 2024. |
| Maine Geological Survey – Maine Cooperative Snow Survey Program | MGS | Maine, New Hampshire and transboundary Canadian watersheds | Snow course | SWE, SD. Bulk density is derived from SWE and SD | https://mgs-maine.opendata.arcgis.com/datasets/maine-snow-survey-data/explore accessed January 2021. |
| Northeast Regional Climate Center | NRCC | New York, Vermont | Snow course | SWE, SD. | https://www.nrcc.cornell.edu |

| | | | | Bulk density derived from SWE and SD | Data received via direct email January 2021. |
|---|---|---|---|---|---|
| New Hampshire Department of Environmental Services (DES) – Dams | NHDES | New Hampshire | Snow course | SWE, SD.<br><br>Bulk density derived from SWE and SD | Data for 1950–2020 received via direct email January 2021 Data for 2019-present available from: https://nhdes.rtiamanzi.org/snow_data |
| National Operational Hydrologic Remote Sensing Center (NOHRSC) | NOHRSC | US and southern Canadian prairies | Airborne gamma radiation | SWE | https://www.nohrsc.noaa.gov/snowsurvey/ accessed January 2022. |
| Finish Environmental Institute – SYKE | SYKE | Finland | Snow course | SWE | https://www.syke.fi/en-US/Open_information/Open_web_services/Environmental_data_API#Hydrology |
| All-Russia Research Institute of Hydrometeorological Information – World Data Center – RIHMI-WDC | RIHMI | Russia | Snow course | SWE, SD and bulk snow density. | https://meteo.ru/english/climate/snow1.php accessed June 2021. |
| Norwegian Water Resources and Energy Directorate (NVE) | NVE | Norway and Nepal | Snow course, snow pillows and snow scales, passive gamma radiation sensors, acoustic SD sensors*. | SWE, SD<br><br>Bulk density derived from SWE and SD. | https://hydapi.nve.no/api/v1/ accessed March 2025 Manual snow survey data via direct email February 2025. |
| Swiss Federal Research Institute (WSL) – Institute for Snow and Avalanche Research (SLF): *'GCOS SWE data from 11 stations in Switzerland'* (Marty 2020) | SLF | Switzerland | Single point manual SWE and SD. | SWE, SD<br><br>Bulk density derived from SWE and SD. | https://envidat.ch/#/metadata/gcos-swe-data accessed February 2025. |

*SD measurements from acoustic sensors are included when co-located with an automated SWE sensor.

**Table 2: Description of variables in the NorSWE NetCDF file.**

| Type of variable | Variable name | Description | Dimension | Units |
|---|---|---|---|---|
| Dimension | station_id | Station identification code | station_id | (-) |
| | time | Time | time | Day |
| Observational metadata | lat | Station latitude | station_id | °North |
| | lon | Station longitude | station_id | ° |
| | elevation | Station elevation | station_id | m |
| | source | Data provider | station_id | (-) |
| | station_name | Station name | station_id | (-) |
| | type_mes | Method of measurement for SWE[1] | station_id | (-) |

| | mmask | Mountain mask[2] | station_id | (-) |
|---|---|---|---|---|
| Data | snw | Water equivalent of snow cover (SWE) | station_id, time | kg m$^{-2}$ |
| | snd | Snow depth (SD) | station_id, time | m |
| | den | Bulk snow density | station_id, time | kg m$^{-3}$ |
| Quality control flags | data_flag_snw | Agency data quality flag for SWE[3] | station_id, time | (-) |
| | data_flag_snd | Agency data quality flag for SWE[3] | station_id, time | (-) |
| | qc_flag_snw | NorSWE quality control flag for SWE[4] | station_id, time | (-) |
| | qc_flag_snd | NorSWE quality control flag for SD[4] | station_id, time | (-) |

1: see Table 3
2: see Section 3
3: see Table 5
4: see Table 8


**Table 3: WMO BUFR (WMO, 2019) SWE measurement codes and non-WMO code for airborne gamma SWE.**

| Code | Method of SWE measurement |
|---|---|
| 0 | Multi-point manual snow survey |
| 1 | Single-point manual snow water equivalent measurement |
| 2 | Snow pillow or snow scale |
| 3 | Passive gamma |
| 4 | GNSS/GPS methods |
| 5 | Cosmic ray attenuation |
| 6 | Time domain reflectometry |
| 63 | Missing value |
| 64 | Airborne gamma SWE |
| 7 – 62 | Reserved |

**Table 4: Manual snow survey sampling protocol and equipment. Precise sampling protocols may differ from those described below, especially where multiple entities contribute data to a single agency.**

| Agency or dataset | Survey design | Snow sampler | References |
|---|---|---|---|
| CanSWE | 5–10 snow cylinders and SD along a 150–300 m line representative of the surrounding landcover (Environment Canada, 2004; Brown et al., 2019; Vionnet et al., 2021). Snow cylinders are usually supplemented by 10–15 ruler SD measurements between SWE samples (e.g. double sampling, WMO, 2018).<br><br>Saskatchewan: 3–5 snow cylinders surrounding a single point; measurement location may vary from one measurement date to the next (see *data_flag_snw*, Table 5). | Federal Sampler, ESC-30 Sampler, Prairie sampler (Saskatchewan) | Vionnet et al. (2021) Environment Canada (2004) |
| Northeast Regional Climate Center | 3–5 cylinders surrounding a single general location. Varies by contributing partner organization. | Primarily Adirondak sampler; Federal sampler also used. | Engel et al. (2022) Samantha Borisoff personal communication March 2022 |

| contributing partners | | | |
|---|---|---|---|
| New Hampshire DES | 10 cylinders per site. | Federal Sampler | Engel et al. (2022) Nancy Baillargeon personal communication June 2024 |
| Maine Geological Survey | 10 snow cylinders and SD along a 100 yard (91.44 m) course. | Primarily Federal Sampler, Adirondack sampler also used. | Maine Geological Survey (2016) |
| Finish Environmental Institute | 8 density (snow tube) and 80 snow depth along a 2– 4 km transect. Measurements are distributed according to landcover type. Mean SWE computed by weighted average according to areal land cover percentages. | Korhonen-Melander sampler: 100 cm$^2$ cross sectional area (70 cm long, 10 cm diameter) | Kuusisto (1984); Leppänen et al. (2016); Vershinina (1971, 1974) |
| Russia ROSHYDROMET | Double sampling (WMO, 2018) along a 1 km and 500 m course in open and forested areas, respectively. Open (forest): cores every 200 m (100 m) and SD in between every 20 m (10 m). | VS-43: graduated iron snow cylinder A = 0.005 m$^2$ x 0.6 m long | Bulygina et al. (2011); Haberkorn (2019) |
| NRCS | 5–10 evenly spaced samples along a transect marked by standard snow course markers. | Federal Sampler, McCall cutter | USDA (2012); Fleming et al. (2023) |
| Norwegian Water Resources and Energy Directorate | Several SD and one or more snow cylinders along a fixed snow transect. Sjusjøen – 2 snow cylinders, 10 SD measurements. Atna – five distinct transects at 100 m elevation band increments. Per transect: 2-4 snow cylinders, SD every 50 m. | Sjusjøen: 0.002 m$^3$ steel cylinder Atna: Federal Sampler or 001 m$^3$ or 0.002 m$^3$ steel cylinder (0.0714 m diameter) | Fleig (2013); Stranden and Saloranta (2021); Heidi Stranden personal communication March 2025 |
| Swiss Institute for Snow and Avalanche Research | Single point manual SWE (snow cylinder) from a snow pit within a seasonally fenced measurement field (15m x 15m). SD from a permanently installed aluminum stake within the measurement field. | Graduated aluminium cylinder (ETH cylinder): 0.007 m$^2$ cross-sectional area, 0.55 m long. | Haberkorn (2019); Marty et al. (2023). |


## 2.1 Manual measurements

### 2.1.1 Gravimetric snow surveys

Gravimetric snow surveys, also known as snow courses or snow transects, consist of multiple depth and density measurements collected along a predefined route that are averaged to obtain a single representative SWE value for the entire route (WMO,

2018; **Table 4** and references therein). In general, a double sampling technique is employed where SD and SWE measurements are collected at multiple points (n = ~5–15) along the route with additional SD measurements (n = ~10) collected between these SWE sampling locations. We gathered snow course data from multiple agencies in Canada (consolidated in CanSWE, Vionnet et al. (2021)), the United States, Norway, Finland, and Russia (**Table 1**). **Table 4** provides general sampling

procedures for the contributing datasets; however, even within a given contributing agency, protocols may vary and differ

from those listed. Measurement uncertainty for various snow samplers ranges from ~3% to 13% (Sturm et al., 2010; Stranden and Grønsten, 2011; Table 2 in Dixon and Boon, 2012 and references therein; USDA, 2012; López-Moreno et al., 2020). This value does not include operator error which can increase the uncertainty. **Figure 1** shows the location of the manual snow survey contained in NorSWE.

### 2.1.2 Single point manual measurements – Switzerland

Switzerland maintains a public database of eleven sites with long-term quality-controlled observations and traceable metadata, referred to as the '*GCOS SWE data from 11 stations in Switzerland*' (Marty, 2020). SWE is measured in a snow pit within a seasonally marked measurement field (Haberkorn, 2019; Marty et al., 2023). Measurement uncertainty for various snow samplers as in **Sect. 2.1.1**. However, in deep mountain snowpacks common to the Swiss Alps multiple samples are often

required to cover the full snow depth which can increase the error (López-Moreno et al., 2020; Marty et al., 2023). The reported SD is from a permanent stake within the measurement field and not immediately adjacent to the snow pit which can introduce additional uncertainty to the derived bulk density (Marty et al., 2023). The focus of NorSWE is on snow courses and airborne gamma SWE which are more spatially representative than single point measurements and data from automated sensors which provide detailed, often daily, observations critical for hydrological modelling and forecasting. The Swiss GCOS data do not

meet these criteria but are exceptionally included to help fill a data gap over Europe.

### 2.2 Automated measurements

The automated SWE observations in NorSWE cover Canada, the United States, Norway, and Nepal. When available, we also include snow depth measurements from co-located snow depth sensors.

### 2.2.1 Passive gamma radiation sensors

Measurements from automated passive gamma radiation sensors (GMON, Choquette et al., 2013) deployed in central and eastern Canada by Hydro-Québec, the Government of Newfoundland and Labrador, and Manitoba Hydro were taken directly from CanSWE v6. A passive gamma radiation sensor is also used as the primary instrument at one site in Norway (Breidvatn). GMON sensors relate the attenuation of naturally emitted gamma radiation from the upper layers of the soil to SWE after

accounting for the background soil moisture. Measurement footprint is ~50–100m$^2$ and measurement range is 0–600 mm; readings are typically taken every 6 hrs (distributed hourly). The stated accuracy of common automated passive gamma instruments is $\pm$ 15 mm up to 300 mm and 15% from 300 mm to 600 mm (Campbell Scientific, 2017) but can be as low as 5% with careful site-specific calibration (see Royer et al., 2021 and references therein). When deployed in the field, measurement uncertainty varies according to environmental factors and soil moisture conditions and may exceed the manufacturer

specifications (Stranden et al., 2015; Smith et al., 2017; Royer et al., 2021). The GMON sensors deployed by Hydro Québec

and the Norwegian Water Resources and Energy Directorate (NVE) also have a co-located hourly-recording sonic snow depth sensor.

### 2.2.2 Snow pillows and snow scales

The snow pillow and snow scale data cover western North America, Norway and Nepal (**Fig. 1**) and were obtained from CanSWE v6, the US SNOTEL network, and NVE (**Table 1**). Snow pillows measure SWE from the overlying hydrostatic pressure on a bladder filled with anti-freeze (Beaumont, 1965). Measurement footprint and instrument accuracy vary by model but are generally around 6 to 9 $m^2$ and $\sim \pm 4\%$, respectively (Stranden et al., 2015; USDA, 2012). Snow pillows are prone to errors (both over and underestimates) when the temperature at the ground-snow interface is at the melting point (Johnson and

Marks, 2004). The SNOTEL and NVE are equipped with a co-located acoustic snow depth sensor as are roughly half of the Canadian sites. Compiled and quality-controlled snow pillow datasets for western North America that aim to address biases between accumulated precipitation and SWE (Meyer et al., 2021) are available elsewhere (e.g. Yan et al., 2018; Sun et al., 2019; Musselman, 2021). Due to differing QC procedures (see **Sect. 4**) the snow pillow data in NorSWE may differ slightly from those contained in these other datasets. In many regions, snow pillows are gradually being replaced with the more

environmentally friendly snow scales (Stranden et al., 2024). NorSWE does not differentiate between these two instruments because they are not differentiated in the WMO BUFR codes (WMO, 2019; **Table 3**).

### 2.3 Airborne gamma SWE

NorSWE includes SWE estimates from NOAA's National Operational Hydrologic Remote Sensing Center (NOHRSC) snow survey program (https://www.nohrsc.noaa.gov/snowsurvey/; Carroll, 2001). This network consists of approximately 2,400

flight lines in 25 US states and seven Canadian provinces (Carroll, 2001) as shown in **Fig. 1**. Flight lines are 10–15 km long and 300 m wide. Surveys are conducted once per year near peak SWE, with occasional flights added to capture hydrologically important conditions. The method, which is limited to $\sim$ < 1000 mm SWE, relates the attenuation of gamma radiation emitted from the upper ~20 cm of the soil by the water mass of the snowpack (liquid or solid phase) after accounting for the background soil moisture (Carroll, 2001). Snow-free radiation and soil moisture conditions are obtained from a snow-free flight, usually

conducted in the fall. In the absence of a fall flight a subjective estimates (SE), a default value (DV, typically 35%), or other ground-based measurement (GM or GI) of soil moisture content is used. Error simulations and comparisons with coincident ground-based observations have reported accuracies of 4% to 10% in prairie and agricultural environments (Carroll et al., 1983) and up to ~12% in forested areas (Carroll and Vose, 1984; Vogel, 1985; Carroll and Carroll, 1989a), although some studies have reported larger errors (Glynn, 1988; Figure 9 in Cho et al., 2020a). A comprehensive accuracy assessment of

NOHRSC airborne gamma SWE showed strong correlation with the University of Arizona SWE product across all land covers and forest fractions (Cho et al., 2020b). Underestimation often occurs when there is significant SWE variability along a flight line (Cork and Loijens, 1980; Carroll and Carroll, 1989b). Inaccurate characterization of the soil moisture, often due to changes in the soil moisture after the fall reference flight, is a common source of error (Carroll and Carroll, 1989b; Cho et al., 2020a).

Other known sources of error include biomass, rock outcrops, navigation, and gamma count statistics (Glynn et al., 1988; Cork

and Loijens, 1980; Carroll and Carroll, 1989b). The airborne gamma SWE data does not include coincident observations of

snow depth.

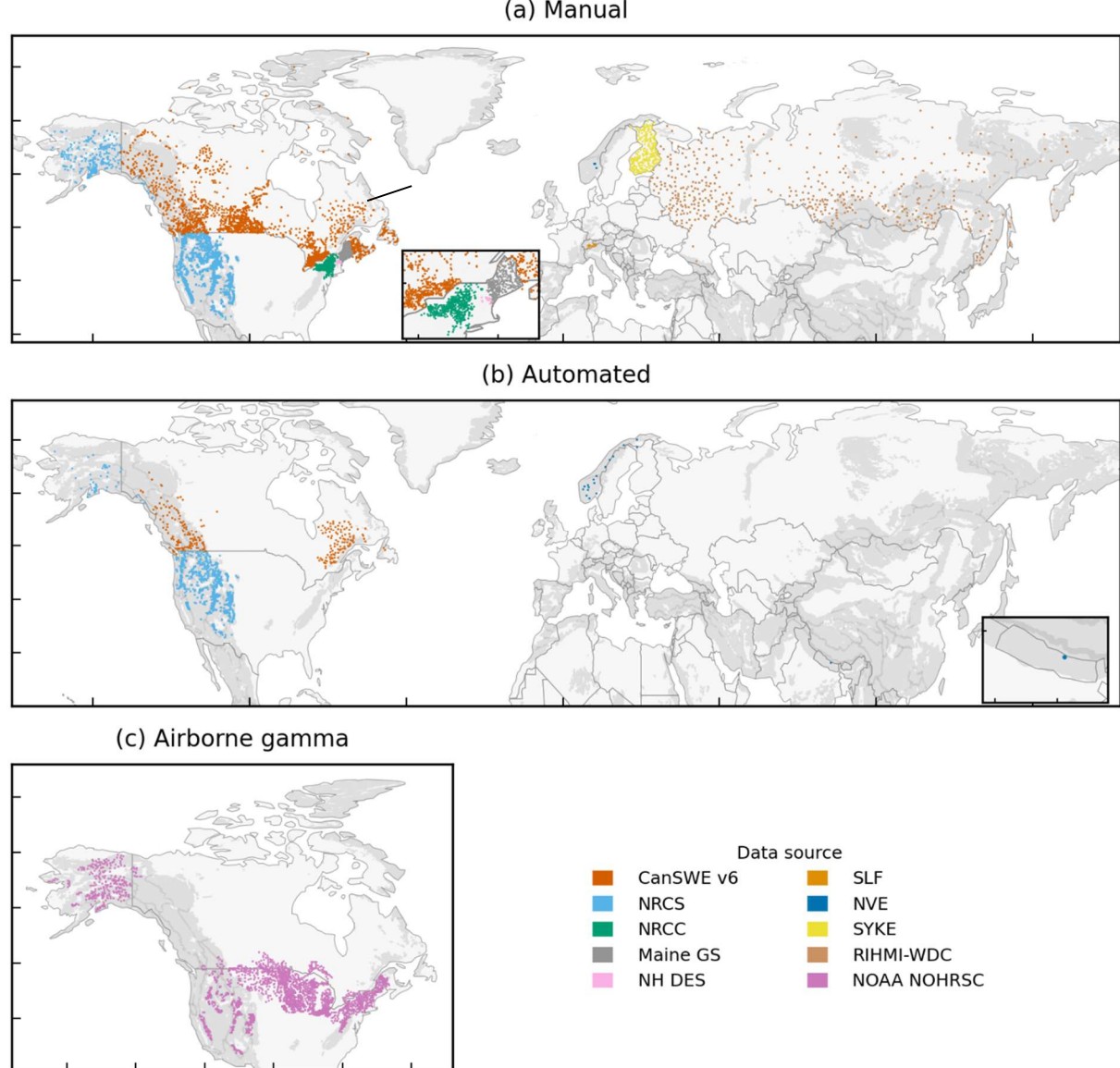

**Figure 1: Spatial distribution of NorSWE sites by measurement type (manual, automated, airborne) and data source. Grey shading indicates mountain mask as detailed in Sect. 3.0.**

## 3 Data cleaning and formatting

Data from each source listed in **Table 1** were obtained either through direct download or email exchange. Data processing followed the steps shown in **Figure 2**. Data cleaning involved removing duplicate stations and observations, correcting obvious errors in measurement dates and removing records flagged as erroneous, adding a mountain classification, and finally converting to a standard netCDF format. Sites intersecting either a 2° slope mask derived from the GETASSE30 DEM or with the Global Mountain Biodiversity Assessment (GMBA) Mountain Inventory v2 (Snethlage et al., 2022; 2023; https://www.earthenv.org/mountains) with a 25 km buffer were flagged as mountain. This broad mountain classification, which is used during quality control (**Sect. 4**), is consistent with that applied in Mortimer et al. (2024) and Mudryk et al. (2025). Data harmonization involved converting imperial units to metric, harmonizing agency-specific quality flags, applying a consistent quality control, checking for duplicate sites between agencies, and finally merging the datasets into a single NetCDF file.

Data for the agencies listed in **Table 1** were cleaned and reformatted to a modified version of the CanSWE NetCDF (**Table 2**). Station metadata include a unique station ID, station name, coordinates consisting of a single latitude, longitude, and elevation, the data source, measurement method and mountain flag (**Table 2**). Station IDs were constructed by prepending the source abbreviation listed in **Table 1** to the original station ID, Russia excepted (see **Sect. 3.2** and **Table 6**). Where elevations were not provided for a given site or network (e.g. Russia and Finland) they were obtained from the USGS' National Elevation Dataset (Gesch et al., 2002). The primary snow variable of interest is SWE. SD and derived bulk snow density, calculated from SD and SWE, are provided when available. Each site, identified by a unique station ID, is permitted only one set of snow observations (snw/snd/den) per day; duplicate observations were removed during data processing (**Sect. 5**). The minimum time resolution in NorSWE is daily.

NorSWE includes two types of flags describing the data quality: agency quality flags and QC flags. QC flags indicate where an observation did not pass our internal quality control (See **Sect. 4**) and was set to *NaN*. Agency quality flags (**Table 5**) incorporate information from the data provider. These can be flags assigned by the agency to indicate certain snow conditions, for example patchy or trace amounts of snow, or to flag observations that were modified or removed during an agency's QC procedures, for example revised data ('R'). Some agencies include one or more comments along with each observation instead of data flags. We coded these comments into flag values using keywords and phrases. For example, records with comments 'skiff' or 'patchy' were assigned a 'T' flag. An exception to the use of the agency quality flag variable is the airborne gamma SWE data which did not have corresponding agency quality information. Instead, we use this variable to store information about the soil moisture estimation method (**Sect. 3.6**).

Agency-specific processing steps are described below except for Finland (SYKE) and Switzerland (SLF) which did not require additional processing beyond the general steps outlined above.

**Table 5: Data flags in NorSWE. Not all data flags are used by all data sources.**

| Data flag | Definition | Comment |
|-----------|------------|---------|

| | | |
|---|---|---|
| A | Sampling problems | |
| B | Manual snow survey conducted outside the nominal sampling period | |
| C | Combination of A and B | |
| E | Estimate | |
| G | Measurement location > 1 km from station coordinate. | Specific to Saskatchewan Water Security Agency housed in CanSWE beginning in 2011. |
| M | Missing | |
| R | Revised data | |
| T | Trace | Includes patchy snow conditions |
| Y | Precise sampling date not available. CanSWE: NWT set to 1 April, within 1 week for Government of Manitoba, research sites (UU) approximate date. NRCS: set to nominal survey date. | |
| AI | Soil moisture – airborne | Data flags for airborne gamma are used |
| AM | Soil moisture – airborne | to store the soil moisture estimation |
| GI | Soil moisture – ground-based information | method. |
| SE | Soil moisture - subjective estimate | |
| MM-avgX | Average of 'X' SWE observations using soil moisture method 'MM' where 'MM' is the soil moisture method AI, AM, GI or SE. | Specific to airborne gamma SWE Lines with >1 observation on a given date, see Section 3.6. |

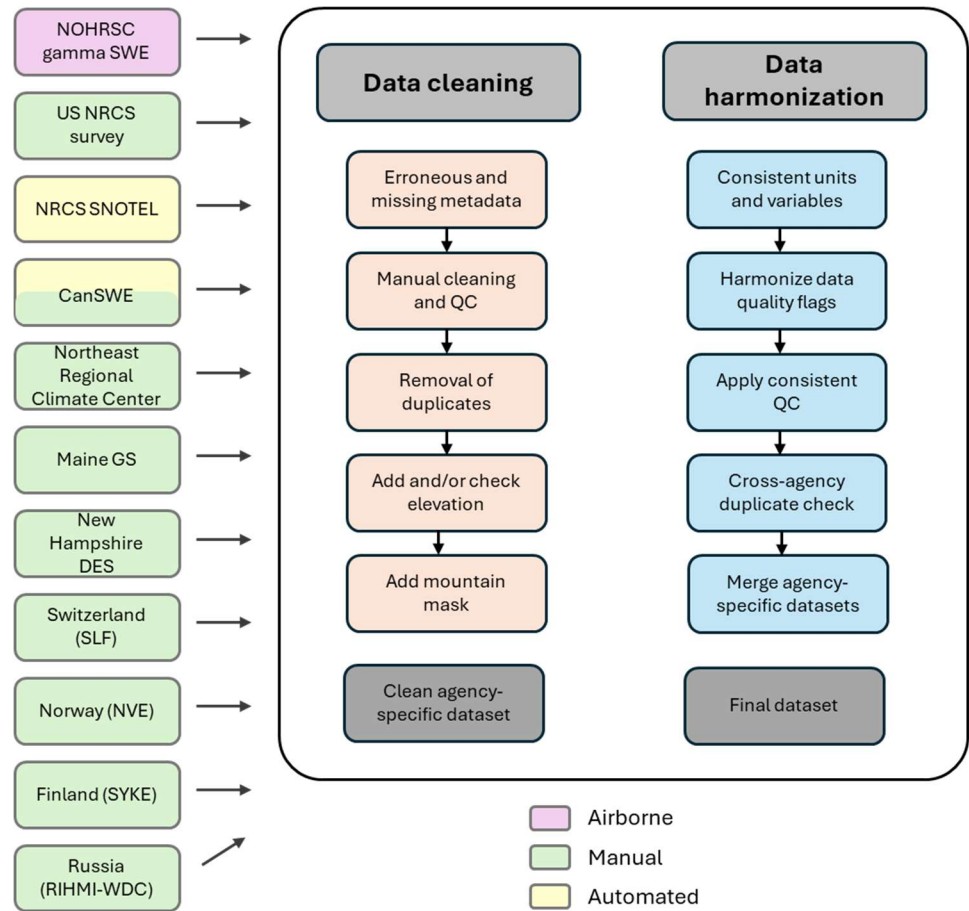

**Figure 2: Schematic of data processing steps. See tables 1 and 2 for information on the contributing datasets listed on the left-hand side.**

### 3.1 Canada (CanSWE)

The CanSWE v6 dataset is included *'as is'* except that we removed the secondary and tertiary station names and IDs to simplify the dataset and added a mountain variable (see **Sect. 4**). Further, we prepended CanSWE to the original station ID and

CanSWEv6 to the original source variable. In CanSWE, only one automated observation per day corresponding to 18:00 UTC is included. In NorSWE, the 18:00 UTC timestamp was dropped to conform to the minimum daily time resolution (**Sect. 3.0**). As described in Vionnet et al. (2021), when only hourly data are available (British Columbia Ministry of Environment and Hydro-Québec) we apply a 24-hour median filter (Stone, 1995) and then extract the record corresponding to 18:00 UTC.

### 3.2 Russia (RIHMI-WDC)

RIHMI-WDC assigns the same WMO ID to up to three distinct snow courses covering different land covers (field/open – 1, forest – 2, and 3 gulley – 3; Bulygina et al., 2011). These distinct snow courses have different sampling frequencies depending

on the land cover. Prior to the spring melt period, sampling of field/open sites is conducted every 10 days when at least half of the visible area is snow covered. Forest sites are sampled once per month prior to 20 January and every 10 days thereafter (**Table 4**, Bulygina et al., 2011). Measurement frequency is every 5 days during spring snowmelt regardless of land cover type.

Land cover type is provided by RIHMI-WDC as a separate variable which is not included in the NorSWE NetCDF format (**Table 2**). To maintain these distinct snow courses while conforming to our NetCDF format we generated new station IDs by appending the landcover flag to the WMO ID as demonstrated in **Table 6**.

**Table 6. Station ID construction for RIHMI-WDC (Russia) for example site 22127 (Lovozero). Not all sites have snow courses for all three land cover types. Station coordinates are the same for each snow course although transects and sampling frequencies differ (Sect. 3.2).**

| Agency prefix (see Table 1) | WMO ID (RIHMI-WDC) | Landcover code | Landcover definition | New NorSWE ID (RIHMI + WMO ID + Landcover code) |
|---|---|---|---|---|
| RHIMI | 22127 | 1 | Field/open | RIHMI-22127_1 |
| RHIMI | 22127 | 2 | Forest | RIHMI-22127_2 |
| RHIMI | 22127 | 3 | Gulley | RIHMI-22127_3 |

### 3.3 Northeast US (Maine GS, NH-DES, NRCC)

Data covering the US northeast were obtained from three sources: Maine Geological Survey (Maine GS), New Hampshire
Department of Environmental Services (NH-DES), and the Northeast Regional Climate Centre (NRCC). Each required substantive manual cleaning to remove duplicate records, inaccurate dates, and to correct errors in the metadata. Sites with missing coordinates were dropped as were records flagged as erroneous by the providing agency. For Maine specifically, records with confidence level marked as '*questionable*' (n = 302) or '*dummy site*' (n = 852) were removed as were those with error codes '*inconsistent or duplicate data within 1, 2, or 3 days*' (n = 156). To avoid creating duplicate records, we removed
all sites from Maine GS with source or station IDs containing Québec, New Brunswick or variations thereof (n = 1392). Quality flags accompanying the original data were retained and harmonized to **Table 5**.

### 3.4 Western US and Alaska (NRCS)

Snow survey data from the US Natural Resources Conservation Service were obtained directly using the GitHub repository https://github.com/CH-Earth/snowcourse. NRCS data compiles observations from state-level data collection offices across the
western US and Alaska. Some states such as California (https://cdec.water.ca.gov/snow.html) also provide these observations through their own data portals. To ensure broad consistency across the region and to avoid introducing duplicate records we chose to draw only from the NRCS database. Records missing exact dates were assigned the date of the nominal survey period and a 'Y' quality flag assigned (**Table 5**). In some remote areas of the western US that are challenging to access the NRCS uses aerial markers instead of snow courses. The marker consists of a large vertical mast (typically a pipe) with horizontal
cross bars that can be seen from a survey aircraft. SD is observed during flyover and SWE is calculated using an estimated

snow density. As these are not direct observations of SWE, we exclude them from NorSWE. Aerial markers were identified as sites with IDs ending in AM or containing AERIAL. This approach, which removed 127 sites, does not account for changes in measurement method over time – sites classified as being aerial markers may include some snow course observations and vice versa. Automated data from the SNOTEL network were obtained using https://ncss-tech.github.io/AQP/soilDB/fetchSCAN-demo.html (Beaudette et al., 2024). Daily SD and SWE downloaded separately and merged by the station ID.

**3.5 Norway (NVE)**

Daily snow observations and metadata for active automated stations managed by the Norwegian Water Resources and Energy Directorate, including one located in Nepal, were obtained using the NVE Hydrological API (https://hydapi.nve.no). Discontinued sites were excluded because these sites were usually abandoned due to measurement issues (Stranden et al., 2015). The SWE and SD observations are daily averages timestamped to 11:00 UTC. The 11:00 UTC timestamp was dropped to conform with the minimum time resolution of NorSWE (**Sect. 3.0**). NVE correction flags were assigned an 'R' flag (revised data, **Table 5**) except for data marked as *interpolated* which were dropped. We include all data regardless of NVE quality flag (unknown, primary controlled, secondary controlled) but use the daily data rather than hourly because they undergo a higher level of quality control. Manual snow survey data were obtained for two locations – Atna and Sjusjøen. The Atna location has five separate transects at 100 m elevation bands between 800 and 1200 m a.s.l.. Each transect has a unique station ID and coordinates. The manual data did not have accompanying quality information or station elevations. For Atna we use the elevation of the corresponding elevation band; Sjusjøen elevation was taken from USGS National Elevation Dataset (Gesch et al., 2002) (**Sect. 3.0**).

**3.6 US and southern Canada airborne gamma SWE (NOHRSC)**

Metadata for the airborne gamma SWE flight lines were derived from the GIS shapefile *'https://www.nohrsc.noaa.gov/gisdatasets/NOHRSC Flight Lines.shp'* as detailed in **Table 7**. LINE became the station ID and station names were constructed from the LINE and River Basin variables. For coordinates, we used the flight line midpoint provided in the file. We calculated missing midpoint coordinates from the flight line endpoints. NOHRSC assigns SWE estimates at or above the method detection limit of 1000 mm a value of 999.0 mm. We excluded these records (n = 32) from our dataset. Finally, we include information about the soil moisture measurement method in the *data_flag_snw* variable. Typically, NOHRSC provides two separate SWE estimates: one based on the measured or estimated soil moisture conditions and one using a soil moisture value of 35% (Carroll, 2001). We only included SWE calculated using either a measured or an estimated soil moisture. In some cases, multiple SWE values are provided for the same date as updated or new soil moisture conditions become available. When this was the case, we averaged these multiple measurements because the last upload or modification date was not consistently available. These averaged records make up <1% of the airborne gamma SWE records

and are identified by '-avgX' appended to the soil moisture estimation code, where X is the number of records averaged which ranged from two to three. There were no instances of different measurement methods being averaged together.

**Table 7. Derivation of variables in NorSWE from the NOHRSC flight index (https://www.nohrsc.noaa.gov/snowsurvey/fline_index.html). The first two letters of the station ID refer to the state or region.**

| NorSWE NetCDF | NOHRSC flights.shp |
|---|---|
| station_id | NOHRSC[2]-NAME (Flight line index 'LINE') |
| station_name | RIVER_BASI-NAME |
| lat[1] | LAT_MIDPNT |
| lon[1] | LON_MIDPNT |
| data_flag_snw | METHOD |

1: If LAT_MID and LON_MID were missing, latitude and longitude were calculated from the flight line endpoints.

2: NOHRSC prepended to 'NAME'

## 4 Quality control

The level of prior quality control of the included datasets differs by agency. Some agencies apply their own quality checks prior to data distribution (e.g. SNOTEL, Fleming et al., 2023) while others share their data *'as is'*. Even when quality control is applied by the collecting agency the methodology is rarely published, often relies on expert judgement, and may not have been applied consistently throughout the time series. This limits our ability to standardize existing QC approaches across constituent datasets and necessitates the application of our own procedure for the merged dataset. To ensure reproducibility of

the quality control, we chose not to implement procedures that rely on ancillary data such as precipitation and temperature (e.g. Johnson and Marks, 2004; Yan et al., 2018; Brown et al., 2021) and instead apply only self-contained methods. Ancillary data are not always consistently available and can be subject to version changes and updates. We encourage users to conduct additional QC using locally available ancillary data when possible. Measurement uncertainty differs according to sensors type, measurement equipment and operational protocols (**Sect. 2**; López-Moreno et al. 2020; Royer et al., 2021; Beaudoin-Galaise

and Jutras, 2022 and references therein); these differences are not addressed in our QC procedure.

Data from each source were subjected to the quality control described in Vionnet et al. (2021), which itself was adopted from Bratten et al. (1998). The QC consists of range thresholding and, for automated sites, an automated outlier detection. **Table 8** outlines the quality control flags used in NorSWE. The infrequency of snow course and airborne gamma observations render QC methods that require near-continuous time series, such as spike checks and automated outlier detection, useless so only

range thresholding is applied. Ranges for SD, SWE and bulk density are 0–3 m (0–8 m for mountain sites), 0–3000 kg m$^{-2}$ (0–8000 kg m$^{-2}$ for mountain sites), and 25–700 kg m$^{-3}$. This approach differs slightly from that of CanSWE which applies the higher thresholds to sites west of 113°W. This change had no impact on the CanSWE data so, metadata aside (see **Sect. 3.1**), the CanSWE records contained in NorSWE are the same as the original (CanSWE v6). Observations outside of these ranges were set to *NaN* and a QC flag assigned according to **Table 8**. Thresholds were applied to SD and SWE separately. For

example, if a record fell outside the SD range but inside the SWE range only the SD record was set to null and the *qc_flag_snd* flag assigned 'H'. If a record failed the snow density test, both SWE and SD were set to null and a 'D' flag assigned to both *qc_flag_snd* and *qc_flag_snw*.

    The automated data were subject to an additional QC step following Hill et al. (2019) (as described in Vionnet et al., 2021). Spurious SD-SWE pairs were identified via the robust sample Mahalanobis distance (Leys et al., 2018) which is the distance

of a point from mean of a multivariate distribution using the minimum covariance determinant. Further details are provided in Leys et al. (2018), Hill et al. (2019), Vionnet et al. (2021), and references therein. The test requires a multivariate dataset so is only applicable to automated sites with both SD and SWE. We required sites to have at least 20 records (SD-SWE pairs) to run this test. Outliers, defined as the upper 0.001 quantile of a chi-squared distribution with $p$ degrees of freedom, were set to null and a QC flag 'V' assigned to both *qc_flag_snd* and *qc_flag_snw*. This method is reasonable at removing extreme outliers

but it has a tendency to also remove valid data during the snow onset and melt periods.

**Table 8: Quality control flags in NorSWE. *NaN* stands for not a number.**

| QC flag | Definition |
| --- | --- |
| H | SD > 3m (> 8m in mountains). SD set to *NaN*. |
| M | Data masked (set to *NaN*) in the precursor to CanSWE. |
| V | Automatic SD-SWE measurement identified as outlier using robust Mahalanobis distance. SD and SWE set to *NaN*. |
| W | SWE > 3000 kgm-2 (> 8000 kgm$^{-2}$ in mountains). SWE set to *NaN*. |
| D | Derived bulk snow density failed 25–700 kgm$^{-3}$ threshold. SD, SWE and bulk snow density set to *NaN*. |

## 5 Merging the datasets

The cleaned and quality-controlled NetCDFs from each contributing agency were merged into a single file after removing duplicate sites and observations between networks (**Fig. 2**, right-hand column). Duplication of records often occurs when a watershed spans multiple jurisdictions (for example southern Canada and northern USA), and data are shared between agencies who each assign their own station IDs.

    Duplicate sites were defined as those with similar locations, snow observations, station names or IDs as follows. First, we

identified all sites from neighbouring agencies with matching station names and inspected those matched sites within 5 km of each other. If the paired sites had matching coordinates and snow records (within rounding precision), we retained the site from the agency whose jurisdiction it intersects. The duplicated site, along with its complete snow record, was dropped. For example, if a site in New Hampshire was found in both the NH-DES and Maine GS databases, we kept the record from NH-DES and dropped the site and its complete observation record from the Maine GS database. Next, from the remaining sites we

identified all sites (same measurement type) within 2 km of any site from a neighbouring agency (e.g. all snow course sites in

CanSWE within 2 km of an NRCS snow course site). The records and station metadata from the matched sites were inspected (compared coordinates, elevations, names, IDs, snow observations) and duplicate sites were dropped. This step removed 63 sites: 56 from CanSWE and 7 from Maine GS (**Table S1**).

The CanSWE v6 dataset with duplicate sites removed and modified metadata (**Sect. 3.1**) was used as the base dataset. The
cleaned datasets from the other ten agencies listed in **Table 1** were added to this base dataset and the time period restricted to 1979–2021.

## 6 Dataset summary

The final dataset contains 11,593,790 valid SWE observations (11,762,366 when including flagged values set to *NaN*) from 10,153 different sites. There are 947,962 SWE observations from 6,672 snow course sites across North America, Norway,
Finland and Russia and an additional 4,000 observations from eleven single point snow pit sites in Switzerland. Together, 2,357 airborne gamma flight lines provide 30,889 SWE observations over the US and southern Canada. Automated sites account for only 11% of the sites in NorSWE but owing to their higher sampling frequency account for 92% of the SWE observations. The 1,113 automated sites are largely restricted to North America, with 17 in Norway and one in Nepal. NorSWE includes 10,700,949 snow pillow observations from 1001 sites across western North America, Norway and Nepal and 134,216
GMON observations from 113 sites in Québec and Newfoundland and Labrador, Canada, and Norway.

Spatially, sites are well distributed across Russia, northern tundra regions excepted, and inland Norway, and there is dense coverage over Finland (**Fig. 1**). In North America, sites are concentrated around populated regions and in the mountain-west but are sparse in the north. This is reflected in the distribution of sites by global seasonal-snow classification (**Fig. 3,** Sturm and Liston, 2021) where there is an over-representation of montane forest, which covers most of the southern populated areas
(**Fig. 3b**), and an under representation of the tundra and boreal forest snow classes whether analysed for North America (Canada and US only, **Fig. S1**) or the complete Northern Hemisphere (**Fig. 3**). Over half of the sites in North America intersect with our mountain mask (**Sect. 3.0**) compared to 15% of Eurasian sites, although all but two Norway sites intersect the mountain mask.

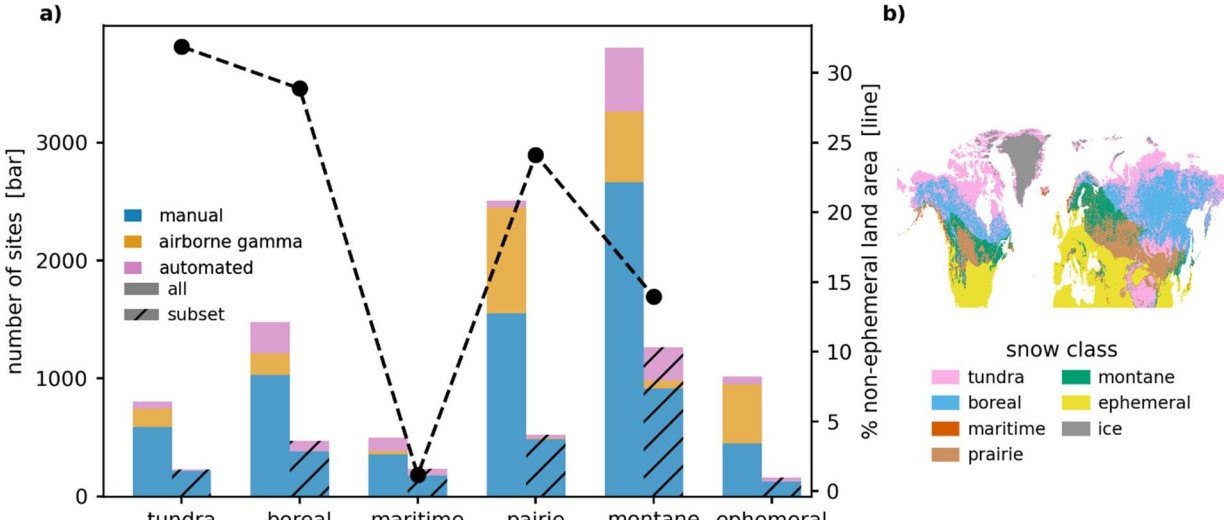

**Figure 3. (a) NorSWE site distribution by snow class (Sturm and Liston, 2021) for the complete dataset (solid bars) and a temporally consistent subset (hatched bars) versus the proportional land area by snow class (dashed black line). The temporally consistent subset consists of sites with at least one measurement in each pentad starting in 1980 and having measurements in at least 30 different years between 1979 and 2021. The ephemeral snow class is excluded from the land area calculations because it does not differentiate between no snow and ephemeral. Permanent land ice is also excluded. (b) Map showing the geographical extent of Sturm and Liston (2021) snow classes. Montane: montane forest, Boreal: boreal forest.**

The number of manual observations in NorSWE has decreased over its time span while the number of automated measurements has increased (**Fig. 4**). While automated instruments provide an alternative to the labour-intensive manual snow courses which can be challenging and costly to conduct in remote locations (Pomeroy and Gray, 1995), the shift away from manual observations can be problematic for the continuity of long-term records without thorough site-specific intercomparisons (e.g. Smith et al., 2017). Further, not all sites in our dataset are sampled consistently throughout its time span. In a given year there are between 3,662 and 5,295 different sites with at least one SWE measurement (**Fig. 4** sum of a and b).

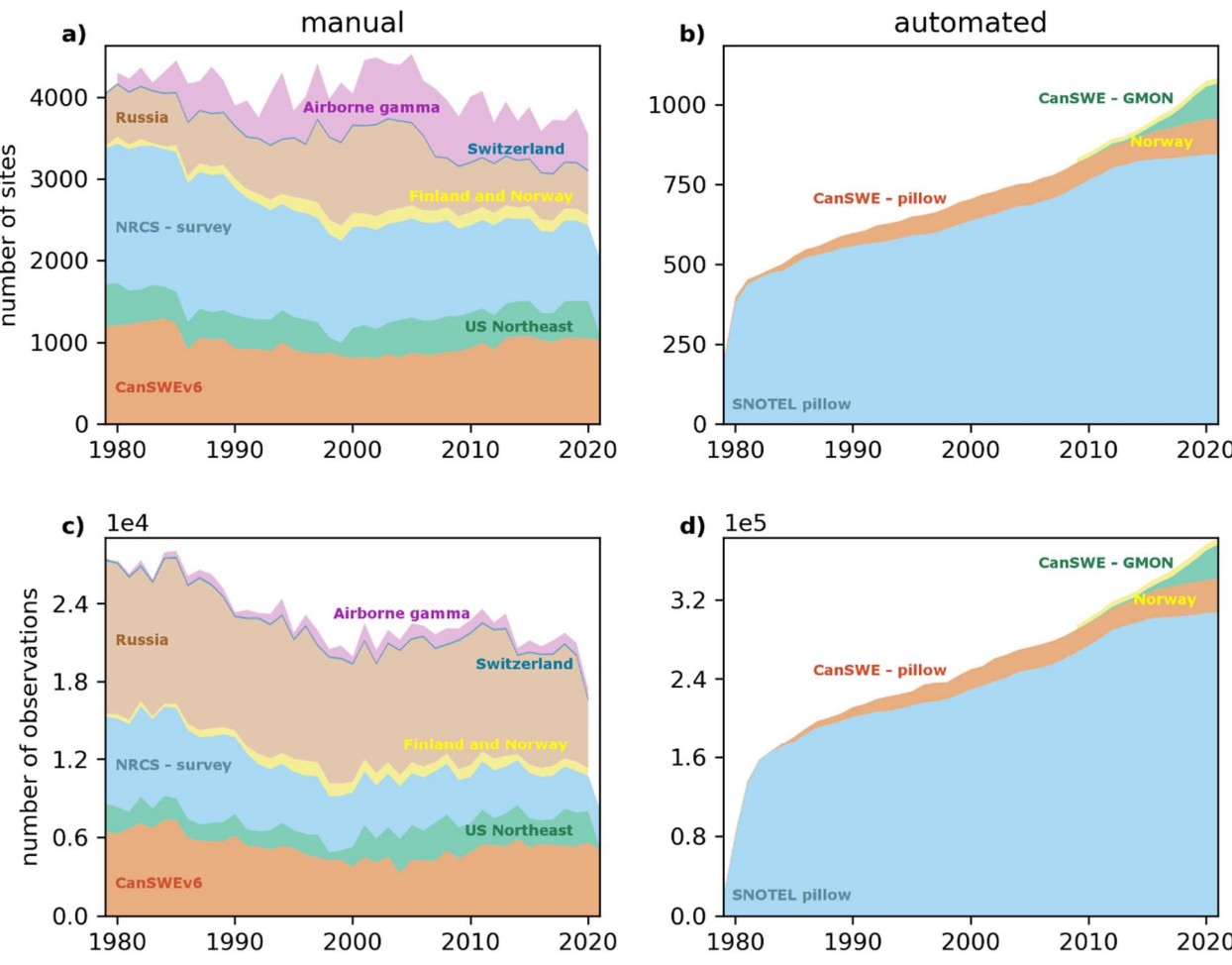

**Figure 4. Number of manual (a and c) and automated (b and d) observations contained in NorSWE. For display purposes, Maine, NH-DES and NRCC combined as are SYKE and the NVE manual sites. Russia typically has 1-3 snow courses for a given site covering different landcovers which are counted separately in NorSWE (Sect. 3.2). Approximately half of the SNOTEL and ~1/3 of the CanSWE and NVE automated SWE values are 0 mm.**

To understand the distribution of consistently sampled sites in NorSWE we identified sites with at least one measurement in each pentad starting in 1980 and having measurements in at least 30 different years between 1979 and 2021 (**Fig. 5**). The SLF, SNOTEL, RIHMI-WDC and NH-DES networks are the most consistent over time with at least half of their sites retained. Similar to the full suite of sites, the temporally consistent subset has well distributed coverage over Finland and Russia, northern regions excepted. In North America, although density of the temporally consistent sites is lower compared to the full complement, coverage remains good in the east and west, maritime Canada excepted. Critically, however, there are almost no sites with consistent long-term (> 30 yrs) records in the central prairies resulting in an increased underrepresentation of the

prairie snow class (**Fig. 3 hatched bars, Fig. S1 and S2**). The lack of North American prairie sites is largely attributed to the curtailment of Canada's ground observing networks starting in the 1980s and to the cessation of the Meteorological Survey of

Canada (MSC) annual Snow Cover Data (SCD) summaries program in 1985 (Brown et al., 2000), combined with inconsistent airborne gamma surveys (**Fig. S3**). The end of the MSC SCD program, which compiled coordinated SWE observations from agencies across Canada, resulted in the loss of historical data collected after this date and before CanSWE's precursor the Canadian Historical Snow Survey Dataset (Brown et al., 2019). Only four Norwegian sites were retained because most NVE sites have only been operational since the late 1990s. The airborne gamma SWE network has the smallest proportion of long-

term consistent sites with only 3% of flight lines meeting our criteria. Airborne gamma observations in Alaska only began in 2003 and flights in much of the western US mountains ceased in the late 2000s and early 2010s (**Fig. S3**).

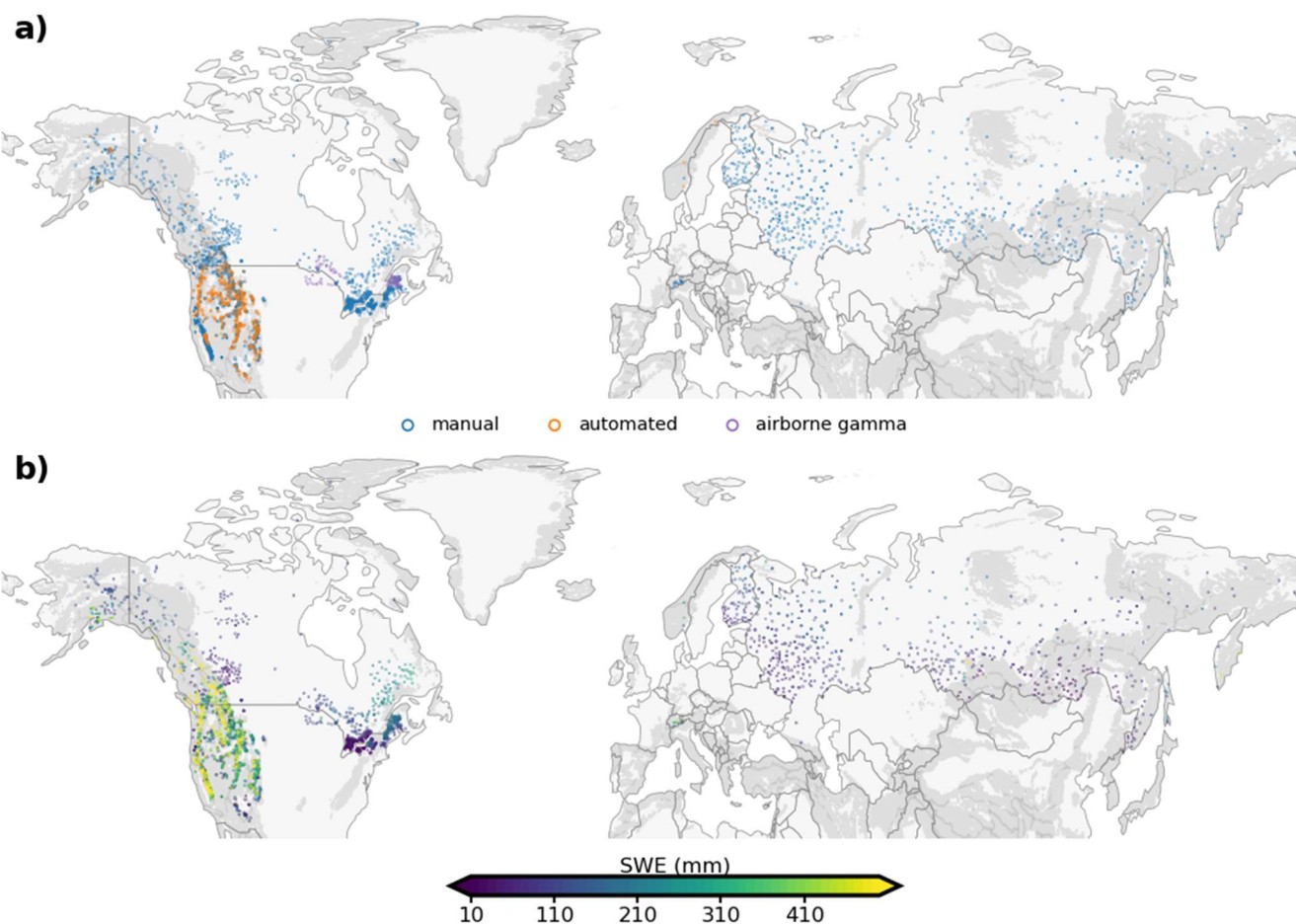

**Figure 5. Sites in NorSWE with at least 30 years of observations and at least 1 year in each pentad starting in 1980 by measurement**
**type (a). (b) mean March SWE for the sites in the top row.**

Lastly, to understand the station representativity according to elevation, we compare site elevations with snow class (ephemeral excluded) as well as mountain and non-mountain landmass hypsometries (**Fig. 6**). The ephemeral snow class is excluded because it includes both ephemeral and no-snow areas. The hypsometries were derived by intersecting the Global Seasonal-Snow Classification, v1 30 arc second, (Liston and Sturm, 2021) reprojected to EASE2 Grid with 1 km spacing, with the Copernicus GLO-30 DEM (https://doi.org/10.5270/ESA-c5d3d65). The elevation distribution of non-mountain sites (**Fig. 6f**) matches that of the terrain, but the elevation distribution of the mountain sites is biased high. The latter reflects that sites in mountain areas tend to be located in the upper reaches of headwater catchments to provide the necessary information for various operational activities. It may also partially reflect the broad mountain mask used in NorSWE (**Sect. 3.0**) where the 25 km buffer applied to the GMBA Mountain Inventory v2 can extend into lower elevation non-mountain areas. Despite the mountain sites being biased high compared to the mountain area hypsometry, they still miss the highest elevation terrain.

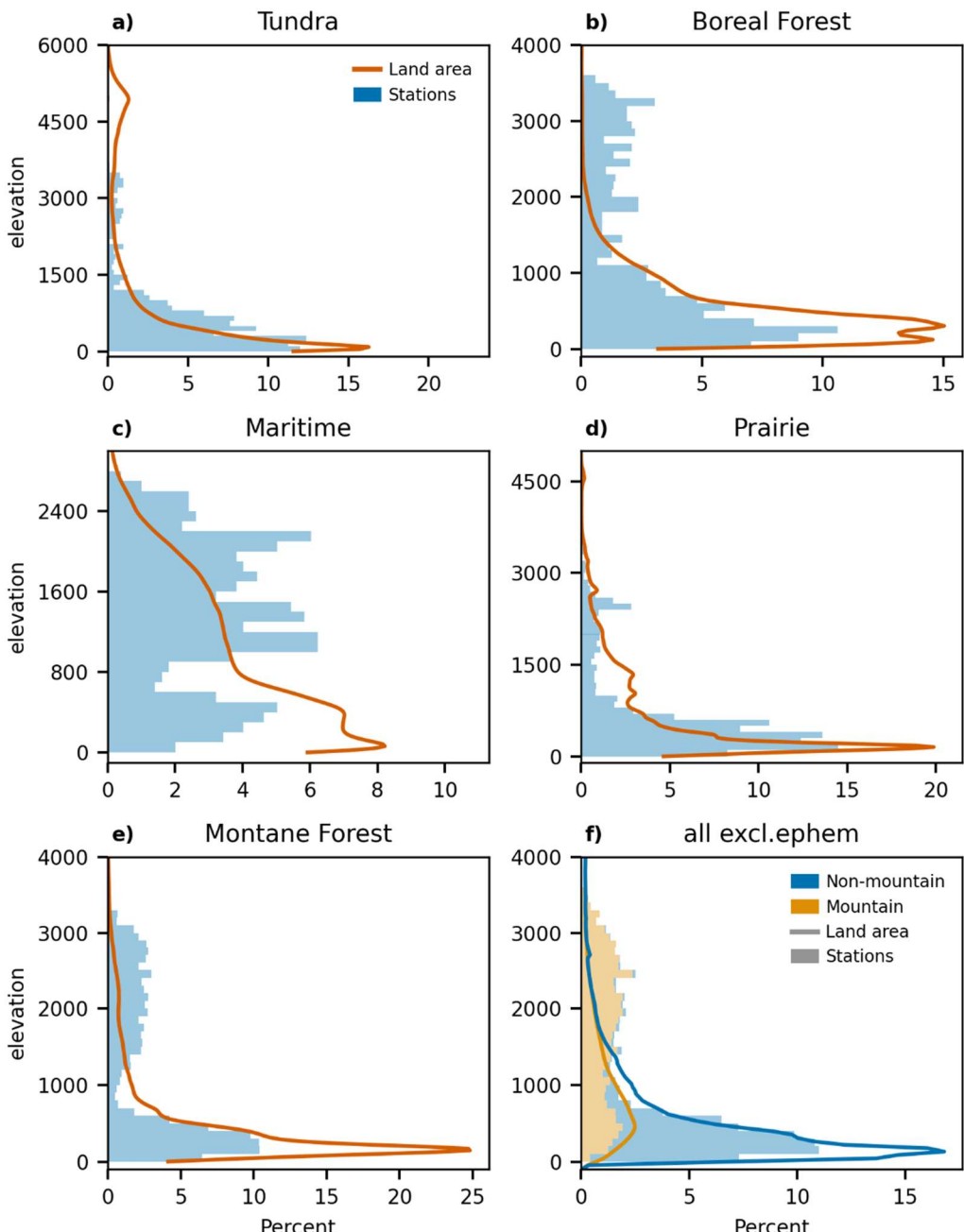

**Figure 6. Elevation distribution of NorSWE sites (blue bars) and land area north of 30°N (orange line) by Sturm and Liston (2021)**
**snow class (a-e). f shows the distribution of sites (bars) and land area north of 30°N (lines) of the five snow classes shown in a-e separated into mountain and non-mountain (Sect. 3.0).**

By snow class, there is representative elevation sampling of the prairie and tundra snow classes, except for the highest elevations of the tundra class. Much of the unsampled high elevation tundra snow is found in High Mountain Asia and in the

mountains of the Canadian Arctic Archipelago where publicly available in situ snow information is lacking. The elevation
distributions of sites in the boreal forest, montane forest, and maritime snow classes are biased high compared to the average terrain. In these snow classes there are generally two peaks: a larger one centred around 200–500 m which aligns with the snow class and is consistent with the non-mountain hypsometry and site distribution (**Fig. 6f**) and a smaller one between 2000 and 3500 m. The second peak, which does not align with the snow class elevation distribution, mirrors the distribution of mountain sites (e.g. **Fig. 6f**). This pattern is reversed for maritime snow.

**7 Dataset usage**

Repeated in situ SWE measurements, such as those contained in our dataset, are critical to understand current climate, state and trends. SWE time series from manual snow courses (see **Table A1** for uses of CanSWE and NorSWE), such as those contained in our dataset, have been used to quantify changes in snow water storage and SWE both regionally (e.g. Bulygina et al., 2011; Hale et al., 2023) and on a hemispheric scale (e.g. Gottlieb and Mankin, 2024) and to tie model-based trends to a
ground truth (Mudryk et al., 2025). They have also been used to benchmark SWE and/or density estimates from satellite data (Luojus et al., 2021; Mortimer et al., 2020; 2022; Gao et al., 2023), reanalysis products and climate models (Mortimer et al., 2020; Elias Chereque et al., 2024; Mudryk et al., 2025), and to understand their uncertainties (Pokorny et al., 2023). Snow density information from a version of our dataset is used to parameterize spatially and temporally varying snow densities applied within the satellite-based GlobSnow and SnowCCI SWE algorithms (Venäläinen et al., 2021; 2023). Automated data
from the well-known and easily accessible SNOTEL network is used extensively as detailed in Fleming et al. (2023). Beyond those data, automated CanSWE data have been used to evaluate hydrological and snowpack models (e.g. Garnaud et al., 2022; Vionnet et al., 2022; Arnal et al., 2024; Marsh et al., 2024), and to validate SWE reconstructions (Sun et al., 2024).

Further, paired SWE-SD measurements are needed to train and validate snow density models used to convert the more plentiful SD observations to SWE (e.g. Sturm et al., 2010; Hill et al., 2019; Sturm and Liston, 2021; Fontrodona-Bach et al., 2023).
Such models, detailed elsewhere (see Avanzi et al., 2015 and references therein; Fontrodona-Bach et al., 2023 and references therein), can fill data gaps by providing estimates of water content when only height is available. However, they require in situ SWE information for their formulation and evaluation, and the quality of these models is strongly tied to the representativeness of available in situ data. For example, density information from CanSWE and SNOTEL were used to train the snow density model used in NH-SWE (Fontrodona-Bach et al., 2023). Comparisons of their model with that from Hill et al. (2019) which
also included in situ data from the US northeast suggests that the inclusion of more data (e.g. NorSWE) would likely improve NH-SWE especially over the northeast US.

In situ SWE observations can also help inform high quality spatially and temporally continuous gridded SWE estimates, exemplified by the University of Arizona SWE product (Broxton et al., 2019). SWE observations from SNOTEL and SD from the US Cooperative network are assimilated with gridded temperature and precipitation (PRISM) data to provide high-quality
daily SWE and SD estimates over the Coterminous US at a 4 km spatial resolution (Zeng et al., 2018). Data contained in

NorSWE could help facilitate the creation of similar datasets at both regional and hemispheric scales. Finally, beyond the snow science community, readily available in situ SWE, SD and snow density information is often required to parameterize physical models (e.g. Dulfer et al., 2022) or as training data for various machine learning applications (Tian et al., 2024).

## 8 Gaps and limitations

NorSWE was initially created to evaluate medium and coarse resolution gridded SWE products (~5–50 km) over the modern satellite period. As such, we focused on snow courses and airborne gamma SWE measurements which are more representative of the surrounding landcover than single point measurements (Meromy et al., 2013). Automated point data were later added to support specific studies (e.g. Arnal et al., 2024).  Although we exceptionally included observations from eleven single point manual sites in Switzerland (**Sect. 2.1.2**; Marty, 2020), our criteria excludes networks that rely primarily on single point SWE

measurements (Haberkorn, 2019). The restricted time period (1979–2021) omits historical data collected before 1979 from NorSWE and it does not extend to present day. Older observations are available from some of the constituent datasets and can be accessed through the links in **Table 1**. Extending our dataset to present day is challenging because, for some agencies, there can be lag of a year or more between data collection and distribution. Another limitation of our dataset is the crude quality control procedure applied to the manual SWE data which relies on common SWE ranges for Canada (Braaten et al., 1998)

which may not be appropriate globally. Further, the infrequent nature of snow course and airborne gamma SWE measurements makes it difficult to apply spike checks and similar procedures to identify erroneous data. Machine learning approaches could be explored to develop improved self-contained QC methods for less frequent observations.

To our knowledge, NorSWE is the most comprehensive in situ SWE dataset for North America covering the modern satellite era. The inclusion of SWE observations from Switzerland, Norway, Finland, Russia, and Nepal provides coverage of most

Northern Hemisphere snow conditions but there are considerable data gaps in south-central Asia and in Europe despite the exception to include the SLF '*GCOS SWE data from 11 stations in Switzerland*' (Marty et al., 2020). Notably, however, these gaps do not necessarily represent an absence of observations. We are aware of networks and sites (see for example Haberkorn et. al., 2019 for Europe, Engel et al., 2022 for the Northeastern U.S.) that were either not made available to the authors or did not meet our initial criteria of long-term snow courses or automated sensors (i.e. they were only operational for a short time

period, provide snow depth and not SWE). Recent initiatives such as the WMO Joint Body for the Status of Mountain Snow Cover   ([https://mountainresearchinitiative.org/flagship-activities/joint-body-on-the-status-of-mountain-snow-cover/](https://mountainresearchinitiative.org/flagship-activities/joint-body-on-the-status-of-mountain-snow-cover/))   may uncover additional sources that could be included in a future version of NorSWE.

## 9 Conclusion

NorSWE (Mortimer and Vionnet, 2025) is a first step towards consolidation and dissemination of in situ SWE data over the

Northern Hemisphere. It combines data from eleven different sources into a single NetCDF file and with a consistent quality

control applied to provide comprehensive coverage of North America as well as Switzerland, Norway, Finland, Russia, and Nepal over the period 1979–2021. It includes both manual and automated observations from four primary methods: snow courses, airborne gamma, snow pillows, and GMON sensors. Single point manual observations over Switzerland are exceptionally included to address data gaps over Europe. Altogether, NorSWE includes 11,593,790 SWE observations from

10,153 different locations. Precursors to this dataset have been used in climate monitoring and research, the development and evaluation of snow products, hydrological modelling, and other activities requiring snow information. NorSWE was possible thanks to the cooperation of individual agencies and to an increase in open data policies. We hope that this dataset will motivate additional agencies to engage in similar data aggregation initiatives.

## 10 Dataset availability

NorSWE is distributed as a single NetCDF file following the Climate and Forecasts (CF) metadata conventions. The NetCDF is distributed as a compressed zipped file (NorSWE-NorEEN_1979-2021_v3.zip) available at https://doi.org/10.5281/zenodo.15263370.

## Acknowledgements

We gratefully acknowledge the field observers collecting manual snow observations and maintaining automatic stations as well as the agencies and personnel who maintaining these records. The authors of NorSWE do not own any of these data. Data are redistributed under the following licences without guarantee of the quality/accuracy of these data. Data from The Norwegian Water Resources and Energy Directorate are distributed, and modified as described herein, under the Norwegian licence for Open Government data (NLOD). CanSWE data are redistributed under the Open Government Licence - Canada.

(https://open.canada.ca/en/open-government-licence-canada). Hydro-Québec data (http://www.hydroquebec.com/) are available under the terms of a Creative Commons Attribution - Non Commercial - Share A Like 4.0 International (https://creativecommons.org/licenses/by-nc-sa/4.0/) (CC BY-NC-SA 4.0). Finnish Environmental Institute (SYKE) data are redistributed under Creative Commons Attribution 4.0 International (CC BY 4.0), 'GCOS SWE data from 11 stations in Switzerland' are redistributed under the Open Database License (ODbL) 'Attribution Share-Alike for Databases'.

The authors of NorSWE have modified the original data as outlined in the manuscript. We recommend users of NorSWE reference the data providers of the relevant agencies as outlined in **Table 4**.

## Appendix A: Uses of CanSWE and NorSWE v1

**Table A1. Demonstration of uses of NorSWE and its precursor CanSWE. NorSWE includes v1 and its unpublished**
**precursors. Uses of Snotel data are detailed in Flemming et al. (2023).**

| Benchmarking gridded SWE products | Dataset |
|---|---|

| | | |
|---|---|---|
| **Mortimer et al. 2022** https://doi.org/10.1016/j.rse.2022.112988 | Benchmarking EO SWE product (Snow CCI+) | NorSWE |
| **Luojus et al. 2021** https://doi.org/10.1038/s41597-021-00939-2 | Validation of GlobSnow v3 product and older CHSSD dataset (Brown et al., 2019) used as input to monthly bias correction. | NorSWE |
| **Gao et al. 2023** https://doi.org/10.3390/rs15082065 | Evaluation of snow densities derived from SMOS over Quebec, Canada. | CanSWE |
| **Mortimer et al. 2024** https://doi.org/10.5194/tc-18-5619-2024 | Impact of in situ method on benchmarking gridded SWE products. | NorSWE |
| **Mudryk et al. 2025** https://doi.org/10.5194/tc-18-5619-2024 | Benchmarking of 23 gridded products from the SnowPex+ intercomparison project. | NorSWE |
| **Elias Chereque et al. 2024** https://doi.org/10.5194/tc-18-4955-2024 | Evaluation of simple temperature index model with different meteorological forcings. | NorSWE |
| **Sun et al. 2024** https://doi.org/10.5194/egusphere-2024-3213 | Evaluation of a mountain SWE reanalysis with snow cover fraction data assimilation. | NorSWE |

**Hydrological model development and evaluation**

| | | |
|---|---|---|
| **Garnaud et al. 2021** https://doi.org/10.3390/rs13245022 | Evaluation of snow analyses in hydrological models for forecasting. | CanSWE |
| **Arnal et al. 2024** https://doi.org/10.5194/hess-28-4127-2024 | Seasonal hydrological forecasting. | NorSWE |
| **Mai et al. 2022** https://doi.org/10.5194/hess-26-3537-2022 | Evaluation and selection of reference datasets for Great Lakes Runoff Intercomparison Project. | CanSWE |
| **Marsh et al. 2024** https://doi.org/10.1029/2023WR036948 | Evaluation of simulated snow drifting patterns with the Canadian Hydrological Model across the Canadian cordillera and adjacent regions. | CanSWE |
| **Shrestha et al. 2022** https://doi.org/10.3389/frwa.2022.801134 | Evaluation of a functional hydrological model of the Great Lakes Basin. | CanSWE |
| **Vionnet et al. 2022** https://doi.org/10.1029/2021WR031778 | Evaluation of the ability of precipitation phase information to improve mountain snowpack prediction. | CanSWE |

**Model input, parameterization, retrieval schemes**

| | | |
|---|---|---|
| **Fontrodona-Bach et al. 2023** https://doi.org/10.5194/essd-15-2577-2023 | CanSWE (and Snotel) data used to develop snow density model to go from SD to SWE. (NH-SWE). | CanSWE |

| Venäläinen et al. 2023<br>https://doi.org/10.5194/tc-17-719-2023 | Interpolated in situ snow density information for use in GlobSnow SWE retrieval. | NorSWE |
| Dulfer et al. 2022<br>https://doi.org/10.1016/j.quascirev.2022.107465 | CanSWE (SD and density) used to calculate snow shielding factors. | CanSWE |
| Tian et al. 2024<br>https://doi.org/10.1016/j.scs.2024.105660 | Training data for machine learning model to investigate the reliability of rapid public transit in the Toronto region under various climate change scenarios. | CanSWE |
| **Snow status and trends** | | |
| Gottlieb and Mankin 2024<br>https://doi.org/10.1038/s41586-023-06794-y | Observational data (CanSWE) used to in attribution study of impact of human influence on NH snow loss. | CanSWE |
| Hale et al. 2023<br>https://doi.org/10.1038/s43247-023-00751-3 | Changes in snow water storage. CanSWE data from 1 April used to evaluate snow storage index output from Snow Storage Index. | CanSWE |
| **Other** | | |
| Pokorny et al 2023<br>https://doi.org/10.1061/JHYEFF.HEENG-5833 | Uncertainty analysis – model uncertainties | CanSWE |

**Competing interests**

The contact author has declared that none of the authors has any competing interests.

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
