# Peer review of "Northern Hemisphere in situ snow water equivalent dataset (NorSWE, 1979-2021)"

_Earth System Science Data, 2024_

## Referee Comment (RC1)

**List of minor comments and technical corrections, with line number (L):**

L 8-9: Revise sentence, I suggest "Here we present the Northern Hemisphere in situ snow water equivalent dataset (NorSWE), consisting …"

L 13: Perhaps a little unimportant, but since you reference the guide to methods and observations from the WMO (2018) in line 25 for the definition of SWE, then I think the acronym HS should be used for snow depth instead of SD throughout the paper and in the dataset, as stated in this same manual.

L 22: I do not understand how the references "National Academies of Sciences, Engineering, and Medicine, 2018; GCOS, 2022" support the statement of "the seasonal snowpack being critical for ecosystems and climate monitoring". Please be more specific or add more relevant references.

L26: Missing a reference for SWE in Global Climate Models.

Caption Table 1: Here you are calling the dataset "NH in situ SWE" instead of "NorSWE". This also occurs on Table 5 and 7, and line 232, and maybe elsewhere, please check and modify.

Table 1: On MGS entry, remove "is" from "Bulk density is derived …" for consistency with other entries.

Table 2: Footnotes (1,2,3) are missing. I guess those would explain the difference between each quality flag?

Table 3: I think "64" should be removed from the caption.

Table 3: What does "7-62 Reserved mean"?

L89: Not sure the abbreviation cf. is properly used here? I had not seen this before, but while searching what it means I found that it is sometimes wrongly used in science. https://scientistseessquirrel.wordpress.com/2016/06/13/friends-dont-let-friends-use-cf/

Lines 94-105: Does this paragraph belong here? It is in section 2.1 on Manual gravimetric snow surveys but the paragraph is about flight surveys and the gamma radiation method.

L 102: What are GM and GI? Should it be "or" instead of "of"?

L 104: Is the uncertainty of the estimates so precise (23 mm)? But I also wonder how reliable these uncertainty ranges are applied to nowadays, considering the papers cited are from 1983 and 1984.

L 114: "can **be** much larger"

Figure 1: add "a,b,c" panels, and perhaps add "Surveys" or "Manual" in panel a?

L 129: "Data from each  source".

L 136: What does "Harmonizing agency-specific quality flags" mean? I got a little confused, because as I understand from Table 2, agency specific quality flags are kept as in the original format in the dataset, in the field "data_flag_snw" and "data_flag_snd", but then in Table 5 the flags are "harmonized", so it is not clear if flags are modified or used as they are. Furthermore, you provide additional quality control flags (Section 4), but these are not specified in Table 2 or 5 (e.g. quality flag "H" in line 235 or "D" in line 236), unless by "CanSWE quality control flag" in Table

2 you mean "NorSWE quality control flag"?. Is there a table or variable missing with your own quality control flags? And how are these distinguished from the original flags from the agencies?

L139: What does "using unique agency-specific Python scripts" mean? Please specify.

L139: coordinate**s**

Table 5: What does "revised data" mean? And what is the difference between Traces and Patches?

Section 3: I find the titles of the subsections a little inconsistent, as some subtitles refer to specific datasets (3.1, 3.2, 3.4), one to a region (3.3) and another one to a method (3.5). I suggest to harmonise them by naming Subsection 3.3 as "Northeast US (MGS, NRCC, NHDES)" and Subsection 3.5 as "Airborne gamma SWE (NOHRSC)", so that at least all datasets are mentioned. Or any other harmonisation that makes the subsectioning clearer.

L148-149: Please explain why SYKE (Finland) did not require any additional processing steps.

L169: Why is the land cover type not supported by NetCDF? Please explain.

Table 6: "Station coordinates  **are** the same ..." and "RIHMI instead of RHIMI.

L 234: I think this should be Table 5 and not 7?

L 243: qg_fla**g**_snd

L 250 and 253: Please specify what "similar coordinates" means.

L 259: "This step removed 63 sites: 62 + 7"? Shouldn't it be 69?

L 264: I suggest to be specific with the exact number of observations in this section, and the exact number of sites.

L 273: I think it should be Fig. 3 or Fig. 5 but not Fig. 4?

Figure 3: I suggest to rephrase the caption, the first three lines are long and without a comma.

L 338: I suggest to remove "briefly", as it is not very brief. Great section though! It is very strong describing so many data usages, well done.

L 344: "uses **to** parameterize"

L 355-357: Yes absolutely! That is why this dataset is great, when I did the data collection myself, I missed these eastern US datasets. Great job!

L 379: Not sure how, but it would be great to include a reference/link for this [https://mountainresearchinitiative.org/flagship-activities/joint-body-on-the-status-of-mountain-snow-cover/](https://mountainresearchinitiative.org/flagship-activities/joint-body-on-the-status-of-mountain-snow-cover/)

Table A1: Perhaps specify for each entry if it is CanSWE or NorSWE that they used? Assuming some of those already used NorSWE. It was not entirely clear to me if the table is about showing examples of "potential uses", or studies that have already used these specific datasets.

---

## Author Comment (AC1)

essd-2024-602 Author response to referee comment 3

Original comments are in black, our responses are in blue, proposed additions and modifications in red. Line numbers refer to the original manuscript.

**Reviewer 3: Alexander Gottlieb**

This paper presents a novel dataset of in situ snow water equivalent (SWE) measurements from North America, Finland, and Russia over the period 1979-2021. The authors clean and harmonize data from 9 distinct sources to provide these measurements in a single file. This represents, to the best of my knowledge, the first publicly-available quasi-hemispheric in situ SWE dataset. As such, I believe it will be of tremendous value to the scientific community and the methods the authors apply seem to have produced a very high-quality dataset. Overall, I would not hesitate to recommend this paper for publication soon. I do, however, have a few minor suggestions that the authors could consider to strengthen the paper even further.

First, while I recognize the tremendous amount of work that goes into wrangling all of this SWE data, there are 2 additional sources that are fairly accessible that could increase the spatial coverage over Europe, particularly in mountainous regions. One is the GCOS dataset from Switzerland, available at https://envidat.ch/#/metadata/gcos-swe-data (citation below) The other is from the Norwegian Water Resources and Energy Directorate: https://www.nve.no/vann-og-vassdrag/vannets-kretsloep/snoe/automatiske-snostasjoner/?ref=mainmenu

Second, the authors do an excellent job of outlining possible uses for their dataset, and there is one more they can consider adding. For applications where spatiotemporally continuous data is necessary, the interpolation of in situ SWE and snow depth data has created some of the highest-quality estimates, such as the 4km University of Arizona SWE product (Broxton et al., 2019). The NorSWE dataset the authors present could facilitate the creation of a high-quality gridded hemispheric products that would be of tremendous value to the research community, and I would encourage them to highlight that as a potential use case.

Overall, I think this is a high-quality dataset that will be very useful, and look forward to the publication of the paper and the use of the data.

Broxton, P., Zeng, X. & Dawson, N. (2019). Daily 4 km Gridded SWE and Snow Depth from Assimilated In-Situ and Modeled Data over the Conterminous US. (NSIDC-0719, Version 1). [Data Set]. Boulder, Colorado USA. NASA National Snow and Ice Data Center Distributed Active Archive Center. https://doi.org/10.5067/0GGPB220EX6A.

Marty, C. (2020). GCOS SWE data from 11 stations in Switzerland. EnviDat. https://www.doi.org/10.16904/15.

Thank you for your constructive feedback and suggestions of additional data and use cases. As suggested also by Adrià Fontrodona-Bach (Reviewer 1) we will add data from the Norwegian Water Resources and Energy Directorate that we missed. For NorSWE, which was originally compiled to evaluate gridded SWE products, we established criteria to only include snow courses and airborne gamma measurements because they are generally more spatially representative than single point

measurements. We later expanded the criteria to also include automated point data because they provide useful information on the seasonal evolution of SWE important for evaluating hydrological models, also an important application of NorSWE data. These criteria omit networks such as the Swiss GCOS data, mentioned by yourself and Adrià Fontrodona-Bach (Reviewer 1), that uses single point measurements. However, due to the significant data gap in Europe and the wide use of the Swiss GCOS data we will make an exception and include these data. The Swiss GCOS sites will be assigned the WMO code 1 (single point manual). Text describing these networks will be added to the revised manuscript and all Figures revised accordingly.

*Marty, Christoph (2020). GCOS SWE data from 11 stations in Switzerland. EnviDat.doi:10.16904/15.*

You make an excellent point about high-resolution gridded datasets that we did not adequately highlight. Indeed, the University of Arizona SWE dataset is an excellent example of this. We propose to add this use case to Section 7.

L349: *"Further, in situ SWE observations can help inform high quality spatially and temporally continuous gridded SWE estimates, exemplified by the University of Arizona SWE product (Broxton et al. 2019). SNOTEL SWE observations and SD from the US Cooperative network are assimilated with gridded temperature and precipitation data to provide high-quality daily SWE and SD estimates over the Coterminous US at a 4km spatial resolution (Zeng et al. 2018). Data contained in NorSWE could help facilitate the creation of similar datasets at both the regional and hemispheric scale."*

Added references

Broxton, P., Zeng, X. and Dawson, N.: Daily 4 km Gridded SWE and Snow Depth from Assimilated In-Situ and Modeled Data over the Conterminous US. (NSIDC-0719, Version 1). [Data Set]. Boulder, Colorado USA. NASA National Snow and Ice Data Center Distributed Active Archive Center. https://doi.org/10.5067/0GGPB220EX6A, 2019.

Zeng, X., Broxton, P., Dawson, N.: Snowpack change from 1982 to 2016 over Conterminous United States, Geophys. Res. Lett., 45,12,940-12,947, https://doi.org/10.1029/2018GL079621, 2018.

---

## Author Comment (AC2)

essd-2024-602 Author response to referee comment 1

Original comments are in black, our responses are in blue, proposed additions and modifications in red, original manuscript text in grey. Line numbers refer to the original manuscript.

**Reviewer 1 Adrià Fontrodona-Bach**

This paper presents a dataset of manual and automated in situ measurements of snow water equivalent (SWE) over the Northern Hemisphere, which is called NorSWE. To the best of my knowledge, there is no in situ global SWE dataset publicly available to date. Currently, researchers need to compile such a dataset from each individual source every time they require it for their applications, and apply their own filters and quality checks, which is time and labour intensive. The scientific community will therefore highly benefit from this dataset, and it fits very well within the scope of the journal. The dataset is excellent, offers a wide range of applications (as the authors very well describe and demonstrate in the paper) and is especially timely as many global products and applications rely on actual SWE measurements and are being increasingly used by the community. The authors did an impressive data compilation and data curation work. The paper is also well written and clear and I hope to see it published soon. However, I have a few minor comments/suggestions and technical corrections that should be addressed before the paper is published.

I only have one rather **major suggestion**, but I call it major just because it may require a bit more time than the rest of minor comments. It regards the spatial coverage of the dataset. The dataset covers a large part of the Northern Hemisphere, but there are some gaps, which the authors recognise in Section 8 (Lines 373-380). It is true that there is a lack of observations in certain areas (e.g. high mountain Asia), and that many other SWE data are just not publicly available. I also acknowledge that it is not possible to find and include every single available dataset and that a line must be drawn somewhere.  However, there is some data available over Europe which the authors did not include and which I strongly encourage they do. These include the Global Climate Observing System (GCOS), the Norwegian Water Resources and Energy Directorate, and a few other individual sites over the Alps. The sources are well listed in Table 1 in Fontrodona-Bach et al. (2023), and they are also used and listed by Seo et al. (2025) in Table 1 https://doi.org/10.5194/essd-2024-349 (preprint). This would give some coverage (despite being still limited) to the European Alps and to Scandinavia outside of Finland, and I strongly encourage that these datasets are included. If the authors wish, they can get in touch with me and I will send them some notes on how I downloaded these data.

Thank you for the constructive feedback. NorSWE was original compiled for the purpose of evaluating gridded SWE products. For that reason, we established criteria to only include snow courses and airborne gamma measurements because they are more spatially representative than single point measurements. We later expanded the criteria to also include automated point data because they provide useful information on the seasonal evolution of SWE important for evaluating hydrological models, also an important application of NorSWE data. However, as you and Alexander Gotlieb (Reviewer 3) point out we missed including data from the Norwegian Water Resources and Energy Directorate. We propose to add these data.

We recognize that our criterion omits certain data, for example single point SWE measurements. The Swiss GCOS network (Marty, 2020), highlighted by yourself and Alexander Gotlieb (Reviewer 3), are single point measurements made at snow pits and therefore do not meet our criteria. However, due to the significant data gap in Europe and the wide use of the Swiss data we will make an exception and include these data. These data will be assigned the WMO code 1 (single point manual).

Text describing these networks will be added to the revised manuscript and all Figures revised accordingly.

*Marty, Christoph (2020). GCOS SWE data from 11 stations in Switzerland. EnviDat.doi:10.16904/15.*

I do not know if the authors requested permission to each individual agency to include their data in NorSWE (if they did, maybe they should explain this in the paper). In any case, in my opinion it is necessary to include clear statements that when using NorSWE data, all the original data sources (so all source datasets) must be appropriately cited as well as the citation of this paper and the NorSWE dataset itself (which is on Zenodo). This provides clear and proper acknowledgement to previous data collections that form this compilations dataset.

We made every attempt to identify the licensing and data redistribution policies associated with each dataset either through direct email with data providers or by investigating the corresponding websites. Indeed, this is the main reason this manuscript and dataset took a long time to come to fruition. Data permissions and redistribution information is included in the NetCDF general attribute 'Distribution'. However, we agree with you that this information is not clearly articulated in our text. Our intent with this dataset is certainly not to take ownership or acknowledgement away from the original data sources. We propose to add an acknowledgements section, which should have been included originally.

*~L396: "11 Acknowledgements*

*We gratefully acknowledge the field observers collecting manual snow observations and maintaining automatic stations as well as the agencies and personnel who maintaining these records. The authors of NorSWE do not own any of these data. Data are redistributed under the following licences without guarantee of the quality/accuracy of the data: NorSWE contains data under the Norwegian licence for Open Government data (NLOD) distributed by The Norwegian Water Resources and Energy Directorate and modified as described herein. CanSWE data are redistributed under the Open Government Licence - Canada. (https://open.canada.ca/en/open-government-licence-canada). Hydro-Quebec data (http://www.hydroquebec.com/) are available under the terms of a Creative Commons Attribution - Non Commercial - Share A Like 4.0 International (https://creativecommons.org/licenses/by-nc-sa/4.0/) (CC BY-NC-SA 4.0). Finnish Environmental Institute (SYKE) data are redistributed under Creative Commons Attribution 4.0 International (CC BY 4.0). US data (NOHRSC, NRCS, Maine Geological Survey, NH DES) are redistributed under the 'US-PD' license (Creative Commons Zero Public Domain Dedication (CC0)), 'GCOS SWE data from 11 stations in Switzerland' are redistributed under the Open Database License (ODbL) 'Attribution Share-Alike for Databases'.*

*The authors of NorSWE have modified the original data as outlined in the manuscript. We recommend users of NorSWE reference the data providers of the relevant agencies as outlined in Table 4."*

I think this dataset will be very useful and I look forward to seeing this paper published and the dataset in use. The rest of comments are listed in the attached pdf.

Thank you for your positive and constructive feedback.

**List of minor comments and technical corrections, with line number (L):**

L 8-9: Revise sentence, I suggest "Here we present the Northern Hemisphere in situ snow water equivalent dataset (NorSWE), consisting ..."
Suggestion implemented.

L 13: Perhaps a little unimportant, but since you reference the guide to methods and observations from the WMO (2018) in line 25 for the definition of SWE, then I think the acronym HS should be used for snow depth instead of SD throughout the paper and in the dataset, as stated in this same manual.
We recognize that the formal WMO acronym for snow depth is HS. However, since SD is more commonly used we will keep it in our paper. Upon the first use of snow depth in the main body of the text we will add explicit mention that HS is the formal abbreviation according to WMO.
*"...situ snow depth (SD), formally abbreviated as HS (WMO, 2018),"*

L 22: I do not understand how the references "National Academies of Sciences, Engineering, and Medicine, 2018; GCOS, 2022" support the statement of "the seasonal snowpack being critical for ecosystems and climate monitoring". Please be more specific or add more relevant references.
We will replace those references with Meredith et al. (2019), Thornton et al. (2021), and Gottlieb and Mankin (2024).

Added references
Thornton, J. M., Palazzi, E., Pepin, N.C., Cristofanelli, P., Essery, R., Kotlarski, S., Giuliani, G., Guigoz, Y., Kulonen, A., Pritchard, D., Li, X., Fowler, H.J., Randin, C.F., Shahgedanova, M., Steinbacher, M., Zebisch M., and Adler, C.: Toward a Definition of Essential Mountain Climate Variables, *One Earth* 4(6): 805–27, https://doi.org/10.1016/j.oneear.2021.05.005, 2021.

Meredith, M., Sommerkorn, M., Cassotta, S., Derksen, C., Ekaykin, A., Hollowed, A., Kofinas, G., Mackintosh, A., J. Melbourne-Thomas, J., Muelbert, M.M.C., Ottersen, G., Pritchard, H. and Schuur, E.A.G.: Polar Regions. In: IPCC Special Report on the Ocean and Cryosphere in a Changing Climate [Pörtner, H.-O., Roberts, D.C., Masson-Delmotte, V., Zhai, P., Tignor, M., Poloczanska, E., Mintenbeck, K., Alegría, A., Nicolai, M., Okem, A., Petzold, J., Rama, B., Weyer, N.M. (eds.)], https://www.ipcc.ch/site/assets/uploads/sites/3/2019/12/SROCC_FullReport_FINAL.pdf, 2019.

L26: Missing a reference for SWE in Global Climate Models.
Will add Mudryk et al. (2020) which is concerned with snow in CMIP6.

Mudryk, L., Santolaria-Otín, M., Krinner, G., Ménégoz, M., Derksen, C., Brutel-Vuilmet, C, Brady, M., and Essery, R.: Historical Northern Hemisphere snow cover trends and projected changes in the CMIP6 multi-model ensemble, The Cryosphere, 14(7), 2495–2514, https://doi.org/10.5194/tc-14-2495-2020, 2020.

Caption Table 1: Here you are calling the dataset "NH in situ SWE" instead of "NorSWE". This also occurs on Table 5 and 7, and line 232, and maybe elsewhere, please check and modify.
Thank you for catching this and we apologize for these sloppy errors. Instances of "NH in situ SWE" replaced with "NorSWE".

Table 1: On MGS entry, remove "is" from "Bulk density is derived …" for consistency with other entries.
Again, thank you for catching this inconsistency. Change implemented.

Table 2: Footnotes (1,2,3) are missing. I guess those would explain the difference between each quality flag?
Footnotes added. We also propose adding text to Section 3 to clarify the difference between agency and QC flags and to provide more information about the agency flags. Table added (Table 6) to describe the QC flags applied immediatly following Table 5 (~L165).

Table 2 footnotes (~L75)
1: see Table 3
2: see Table 5
3: see Table 6

L144: *"Each site, identified by a unique station ID, is permitted only one set of snow observations (snw/snd/den) per day; duplicate observations are removed during data processing. NorSWE includes two types of flags describing the data quality: agency quality flags and qc flags. QC flags indicate where an observation did not pass our quality control (See Sect. 4) and was set to nan. Agency quality flags (Table 5) incorporate information from the original observations from the data provider. These can be flags assigned by the agency to indicate certain snow conditions, for example patchy or wet snow, or to flag observations that were modified or removed during their internal QC procedures, for example revised data. Some agencies include one or more comments along with each observation instead of data flags. We coded these comments into flag values using keywords and phrases. For example, records with comments 'skiff' or 'patchy' were assigned a 'T' flag. An exception to the use of the agency quality flag field is the airborne gamma SWE data which did not have corresponding agency quality information, and we instead use this variable to store information about the soil moisture estimation method (Sect. 3.5)."*

**Table 6: Quality control (QC) flags in NorSWE. NaN stands for not a number.**

| QC flag | Definition |
|---|---|
| H | SD > 3m (> 8m in mountains). SD set to *NaN*. |
| M | Data masked (set to *NaN*) in a previous CHSSD update. |
| V | Automatic SD-SWE measurement identified as outlier using robust Mahalanobis distance. SD and SWE set to *NaN*. |
| W | SWE > 3000 kgm-2 (> 8000 kgm$^{-2}$ in mountains). SWE set to *NaN*. |
| D | Derived bulk snow density failed 25–700 kgm$^{-3}$ threshold. SD, SWE and bulk snow density set to *NaN*. |

Table 3: I think "64" should be removed from the caption.
Table 3 (~L78) caption revised to: *"Table 3: WMO SWE measurement codes (WMO, 2019) and non-WMO code for airborne gamma SWE."*

Table 3: What does "7-62 Reserved mean"?
Table 3, except value 64, is taken directly from the WMO SWE measurement codes (WMO, 2019). Our understanding is that numbers 7 – 62 are reserved for future use – i.e. future methods requiring specific codes. Code 3 'passive gamma' does not differentiate between automated gamma measurements from airborne measurements, but these are very different implementations of 'passive gamma' methods, so we added a 'new' (non-WMO) code for airborne gamma specifically. Because 7 – 62 are 'reserved' and 63 is intended for missing measurement type values we assigned airborne gamma the next available numeric code – 64.

L89: Not sure the abbreviation cf. is properly used here? I had not seen this before, but while searching what it means I found that it is sometimes wrongly used in science.
https://scientistseessquirrel.wordpress.com/2016/06/13/friends-dont-let-friends-use-cf/
'cf.' removed

Lines 94-105: Does this paragraph belong here? It is in section 2.1 on Manual gravimetric snow surveys but the paragraph is about flight surveys and the gamma radiation method.
Indeed, Section 2.2 subheading was missing. Will add subheading *2.2 Airborne gamma SWE*.

L 102: What are GM and GI? Should it be "or" instead of "of"?
Corrected. Should be 'or'.

L 104: Is the uncertainty of the estimates so precise (23 mm)? But I also wonder how reliable these uncertainty ranges are applied to nowadays, considering the papers cited are from 1983 and 1984.

The majority of the literature regarding airborne gamma SWE is from the 1980s. Recent work by Eunsang Cho has demonstrated good agreement with the University of Arizona SWE dataset and also included some comparisons with ground observations. We propose to expand the discussion of uncertainties related to airborne gamma SWE. We acknowledge many of the added references are still quite old.

Expanded text:

L102: *"Error simulations and comparisons with coincident ground-based observations have reported accuracies of 4% to 10% in prairie and agricultural environments (Carroll et al. 1983) and up to ~12% in forested areas (Carroll and Vose, 1984; Vogel, 1985; Carroll and Carroll 1989a), although some studies have reported larger errors Glynn 1988; Cho et al. 2020a Figure 9). A comprehensive accuracy assessment of NOHRSC airborne gamma SWE showed strong correlation with the University of Arizona SWE product across all land covers and forest fractions (Cho et al. 2020b). Underestimation often occurs when there is significant SWE variability along a flight line (Cork and Loijens, 1980; Carroll and Carroll 1989b). Inaccurate characterization of the soil moisture, often due to changes in the soil moisture after the fall reference flight, is a common source of error (Carroll and Carroll 1989b; Cho et al. 2020a). Other known sources of error include*

*biomass, rock outcrops, navigation, and gamma count statistics (Glynn et al. 1988; Cork and Carroll and Carroll 1989a; Carroll and Carroll 1989b)."*

*Added references*

*Carroll, T. R., Glynn, J.E., and Goodison, B.E.: A comparison of U.S. and Canadian airborne gamma radiation snow water equivalent measurements, Proc. West. Snow Conf., 51, 27-37, 1983.*

*Carroll, S.S., Carroll, T.R.: Effect of uneven snow cover on airborne snow water equivalent estimates obtained by measuring terrestrial gamma radiation, Water Resour. Res., 25 (7), 1505 - 1510,* https://doi.org/10.1029/WR025i007p01505, *1989b.*

*Carroll, S.S. and Carroll, T.R.: Effect of forest biomass on airborne snow water equivalent estimates obtained by measuring terrestrial gamma radiation. Remote Sens. Environ. 27 (3), 313 - 319. https://doi.org/10.1016/0034-4257(89)90091-6, 1989a.*

*Cho, E., Jacobs, J.M., Schroeder, R., Tuttle, S. E. and Olheiser, C.: Improvement of operational airborne gamma radiation using SMAP soil moisture, Remote Sens. Environ., 240, 111668, https://doi.org/10.1016/j.rse.2020.111668, 2020a.*

*Cho, E., Jacobs, J.M., and Vuyovich, C.: The value of long-term (40 years) airborne gamma radiation SWE record for evaluating three observation-based gridded SWE datasets by seasonal snow and land cover classifications, Water Resour. Res., 56, e2019WR025813, https://doi.org/10.1029/2019WR025813, 2020b.*

*Cork, H. F. and H. S. Loijens, H.S.: The effect of snow drifting on gamma survey results, J. Hydrol., 48(1-2), 41-51, https://doi.org/10.1016/0022-1694(80)90064-5, 1980.*

*Glynn, J.E., Carroll, T.R., Holman, P.B., Grasty, R.L.: An airborne gamma ray snow survey of a forested covered area with a deep snowpack, Remote Sens. Environ., 26 (2), 149 - 160, https://doi.org/10.1016/0034-4257(88)90093-4, 1980.*

*Vogel, R. M., Carroll, T. R., and Carroll, S. S.: Simulation of airborne snow water equivalent measurement errors made over a forest environment, Proceedings of the American Society of Civil Engineers Symposium, Denver, CO, p. 9, 1985.*

L 114: "can **be** much larger"
Correction made.

Figure 1: add "a,b,c" panels, and perhaps add "Surveys" or "Manual" in panel a?
Will add 'a', 'b', 'c' added to panels. Will rename panel 'a' to 'Manual' and distinguish between surveys and single point (Swiss GCOS).

L 129: "Data from each of source".
Corrected to 'Data from each source listed …'

L 136: What does "Harmonizing agency-specific quality flags" mean? I got a little confused, because as I understand from Table 2, agency specific quality flags are kept as in the original format in the dataset, in the field "data_flag_snw" and "data_flag_snd", but then in Table 5 the flags

are "harmonized", so it is not clear if flags are modified or used as they are. Furthermore, you provide additional quality control flags (Section 4), but these are not specified in Table 2 or 5 (e.g. quality flag "H" in line 235 or "D" in line 236), unless by "CanSWE quality control flag" in Table 2 you mean "NorSWE quality control flag"?. Is there a table or variable missing with your own quality control flags? And how are these distinguished from the original flags from the agencies?

Thank you for this comment. It is clear we did not adequately explain the difference between agency and QC flags, and our text discussing 'agency quality flags' is insufficient.
We propose to add the following text to Section 3 to better provide a better explanation of 'agency quality flags' and to offer better distinction between them and QC flags. We will also add a table describing the QC flags (new Table 6).

L146: "*NorSWE includes two types of flags describing the data quality: agency quality flags and qc flags. QC flags indicate where an observation did not pass our quality control (See Sect. 4) and was set to NaN. Agency quality flags (Table 5) incorporate information concerning the original observation as detailed by the data provider. These can be flags assigned by the agency to indicate certain snow conditions, for example patchy or wet snow, or to flag observations that were modified or removed during their internal QC procedures, for example revised data. Some agencies include one or more comments along with each observation instead of data flags. We coded these comments into flag values using keywords and phrases. For example, records with comments 'skiff' or 'patchy' were assigned a 'T' flag. An exception to the use of the agency quality flag field is the airborne gamma SWE data which did not have corresponding agency quality information; instead we use this variable to store information about the soil moisture estimation method (Sect. 3.5).*"

**Table 6: Quality control (QC) flags in NorSWE. NaN stands for not a number.**

| QC flag | Definition |
|---------|------------|
| H | SD > 3m (> 8m in mountains). SD set to *NaN*. |
| M | Data masked (set to *NaN*) in a previous CHSSD update. |
| V | Automatic SD-SWE measurement identified as outlier using robust Mahalanobis distance. SD and SWE set to *NaN*. |
| W | SWE > 3000 kgm-2 (> 8000 kgm$^{-2}$ in mountains). SWE set to *NaN*. |
| D | Derived bulk snow density failed 25–700 kgm$^{-3}$ threshold. SD, SWE and bulk snow density set to *NaN*. |

L139: What does "using unique agency-specific Python scripts" mean? Please specify.
'using unique agency-specific Python scripts' removed.

L139: coordinate**s**
Corrected

Table 5: What does "revised data" mean? And what is the difference between Traces and Patches?

Thank you for the question regarding 'Trace' versus 'Patches'. We propose to consolidate these into a single 'Trace' flag. The 'Patches' line will be removed from Table 5 and P flags replaced with T flags (697 snow course instances). Revised data means the data were revised by the original providing agency. This is now clarified in the expanded Section 3 text (above), relevant excerpt:
"*...to flag observations that were modified or removed during their internal QC procedures, for example revised data.*"

Section 3: I find the titles of the subsections a little inconsistent, as some subtitles refer to specific datasets (3.1, 3.2, 3.4), one to a region (3.3) and another one to a method (3.5). I suggest to harmonise them by naming Subsection 3.3 as "Northeast US (MGS, NRCC, NHDES)" and Subsection 3.5 as "Airborne gamma SWE (NOHRSC)", so that at least all datasets are mentioned. Or any other harmonisation that makes the subsectioning clearer.

Agreed. We propose the following harmonized subsections:

3.1 Canada (CanSWE)
3.2 Russia (RIHMI-WDC)
3.3 Northeast US (MGS, NHDES, NRCC)
3.4 Western and Alaska US (NRCS)
3.5 Airborne gamma SWE (NOHRSC)

L148-149: Please explain why SYKE (Finland) did not require any additional processing steps.
The data from SYKE was clean and did not include any agency data flags so extra processing steps were required. Only the general processing steps were applied.

L169: Why is the land cover type not supported by NetCDF? Please explain.
Text in question removed.

Table 6: "Station coordinates  **are** the same ..." and "RIHMI instead of RHIMI.
Corrected

L 234: I think this should be Table 5 and not 7?
Revised to refer to new Table 6 (other Tables renumbered accordingly).

**Table 6: Quality control (QC) flags in NorSWE. NaN stands for not a number.**

| QC flag | Definition |
|---------|------------|
| H | SD > 3m (> 8m in mountains). SD set to *NaN*. |
| M | Data masked (set to *NaN*) in a previous CHSSD update. |
| V | Automatic SD-SWE measurement identified as outlier using robust Mahalanobis distance. SD and SWE set to *NaN*. |
| W | SWE > 3000 kgm-2 (> 8000 kgm$^{-2}$ in mountains). SWE set to *NaN*. |
| D | Derived bulk snow density failed 25–700 kgm$^{-3}$ threshold. SD, SWE and bulk snow density set to *NaN*. |

L 243: qg_fladg_snd
Corrected

L 250 and 253: Please specify what "similar coordinates" means.
Text revised for clarity.
L250: *"Duplicate sites were defined as those with similar locations, snow observations, station names or IDs as follows. First, we identified all sites from neighbouring agencies with matching station names and inspected those matched sites within 5 km of each other. If the paired sites had matching coordinates and snow records (within rounding precision), we retained the site from the agency whose jurisdiction it intersects…"*

L 259: "This step removed 63 sites: 62 + 7"? Shouldn't it be 69?

*According to Table S1 it should read "This step removed 63 sites: 56 from CanSWE and 7 from Maine GS (Table S1)."*

L 264: I suggest to be specific with the exact number of observations in this section, and the exact number of sites.

As per your suggestion we will provide exact numbers for the revised dataset.

L 273: I think it should be Fig. 3 or Fig. 5 but not Fig. 4?

Corrected to Fig. 3.

Figure 3: I suggest to rephrase the caption, the first three lines are long and without a comma. Proposed revision:

*Figure 3. **(Left):** NorSWE site distribution by **snow class (Sturm and Liston, 2021)** snow class for the complete dataset **(solid bars)** and a **temporally consistent** subset **(hatched bars)** versus the proportional land area by snow class **(dashed black line)**. **The temporally consistent subset consists of** sites with at least one measurement in each pentad starting in 1980 and having measurements in at least 30 different years between 1979 and 2021. The ephemeral snow class is excluded from the land area calculations because it does not differentiate between no snow and ephemeral. Permanent land ice is also excluded. **(Right): Map showing the geographical extent of Sturm and Liston (2021) snow classes.** Montane: montane forest, Boreal: boreal forest.*

L 338: I suggest to remove "briefly", as it is not very brief. Great section though! It is very strong describing so many data usages, well done.

Suggestion implemented.

L 344: "uses **to** parameterize"

Corrected.

L 355-357: Yes absolutely! That is why this dataset is great, when I did the data collection myself, I missed these eastern US datasets. Great job!

Your dataset is great. We just thought the differences in North America were quite interested. As you know, data collation is challenging and takes a village, so we greatly appreciate your suggestions about additional European data.

L 379: Not sure how, but it would be great to include a reference/link for this https://mountainresearchinitiative.org/flagship-activities/joint-body-on-the-status-of-mountain-snow-cover/

Agreed. We propose to add the link to the activity.

Table A1: Perhaps specify for each entry if it is CanSWE or NorSWE that they used? Assuming some of those already used NorSWE. It was not entirely clear to me if the table is about showing examples of "potential uses", or studies that have already used these specific datasets.

Table A1 is intended to show the ranges of uses of this type of dataset. Title renamed accordingly and dataset used added.

*Table A1: **Demonstration of uses of NorSWE and its precursor CanSWE. NorSWE includes v1 and its unpublished precursors. Uses of Snotel data are detailed in Flemming et al. 2023.***

| Benchmarking gridded SWE products | Dataset |
| --- | --- |

| | | |
|---|---|---|
| **Mortimer et al. 2022** https://doi.org/10.1016/j.rse.2022.112988 | Benchmarking EO SWE product (Snow CCI+) | NorSWE |
| **Luojus et al. 2021** https://doi.org/10.1038/s41597-021-00939-2 | Validation of GlobSnow v3 product and older CHSSD dataset (Brown et al., 2019) used as input to monthly bias correction. | NorSWE |
| **Gao et al. 2023**. https://doi.org/10.3390/rs15082065 | Evaluation of snow densities derived from SMOS over Quebec, Canada. | CanSWE |
| **Mortimer et al. 2024** https://doi.org/10.5194/tc-18-5619-2024 | Impact of in situ method on benchmarking gridded SWE products. | NorSWE |
| **Mudryk et al. 2024** https://doi.org/10.5194/egusphere-2023-3014 | Benchmarking of 23 gridded products from the SnowPex+ intercomparison project. | NorSWE |
| **Elias Chereque et al. 2024** https://doi.org/10.5194/tc-18-4955-2024 | Evaluation of simple temperature index model with different meteorological forcings. | NorSWE |
| **Sun et al. 2024** https://doi.org/10.5194/egusphere-2024-3213 | Evaluation of a mountain SWE reanalysis with snow cover fraction data assimilation. | NorSWE |
| **Hydrological model development and evaluation** | | |
| **Garnaud et al. 2021** https://doi.org/10.3390/rs13245022 | Evaluation of snow analyses in hydrological models for forecasting. | CanSWE |
| **Arnal et al. 2024** https://doi.org/10.5194/hess-28-4127-2024 | Seasonal hydrological forecasting. | NorSWE |
| **Mai et al. 2022** https://doi.org/10.5194/hess-26-3537-2022 | Evaluation and selection of reference datasets for Great Lakes Runoff Intercomparison Project. | CanSWE |
| **Marsh et al. 2024** https://doi.org/10.1029/2023WR036948 | Evaluation of simulated snow drifting patterns with the Canadian Hydrological Model across the Canadian cordillera and adjacent regions. | CanSWE |
| **Shrestha et al. 2022** https://doi.org/10.3389/frwa.2022.801134 | Evaluation of a functional hydrological model of the Great Lakes Basin. | CanSWE |
| **Vionnet et al. 2022** https://doi.org/10.1029/2021WR031778 | Evaluation of the ability of precipitation phase information to improve mountain snowpack prediction. | CanSWE |
| **Model input, parameterization, retrieval schemes** | | |
| **Fontrodona-Bach et al. 2023** https://doi.org/10.5194/essd-15-2577-2023 | CanSWE (and Snotel) data used to develop snow density model to go from SD to SWE. (NH-SWE). | CanSWE |
| **Venäläinen et al. 2023** https://doi.org/10.5194/tc-17-719-2023 | Interpolated in situ snow density information for use in GlobSnow SWE retrieval. | NorSWE |
| **Dulfer et al. 2022** https://doi.org/10.1016/j.quascirev.2022.107465 | CanSWE (SD and density) used to calculate snow shielding factors. | CanSWE |
| **Tian et al. 2024** https://doi.org/10.1016/j.scs.2024.105660 | Training data for machine learning model to investigate the reliability of rapid public transit in the Toronto region under various climate change scenarios. | CanSWE |
| **Snow status and trends** | | |
| **Gottlieb and Mankin 2024** https://doi.org/10.1038/s41586-023-06794-y | Observational data (CanSWE) used to in attribution study of impact of human influence on NH snow loss. | CanSWE |
| **Hale et al. 2023** https://doi.org/10.1038/s43247-023-00751-3 | Changes in snow water storage. CanSWE data from 1 April used to evaluate snow storage index output from Snow Storage Index. | CanSWE |

| Other | |
|---|---|
| **Pokorny et al 2023**
 https://doi.org/10.1061/JHYEFF.HEENG-5833 | Uncertainty analysis – model uncertainties | CanSWE |

---

## Author Comment (AC3)

essd-2024-602 Author response to referee comment 2

Original comments are in black, our responses are in blue, proposed additions and modifications in red, original manuscript text in grey. Line numbers refer to the original manuscript.

**Reviewer 2**

I appreciate the authors' effort in creating a centralized data repository for all snow data. I have a few suggestions to further enhance the paper:

The purpose of the dataset is not a centralized repository for all snow data but rather a compilation of in situ SWE measurements. The dataset was initially compiled to support evaluation of gridded SWE products so was limited to snow course and airborne gamma SWE which are more spatially representative than single point measurements. It was later expanded to include automated SWE measurements covering North America to support evaluation of hydrological models. These criteria exclude derived data and those based on satellite observations. Further we stress that the dataset's focus is on SWE, not snow cover. Snow depth and snow density are only included when available alongside the SWE observations. Consequently, snow depth measurements from automatic stations, non collocated to SWE measurements, which are collected by many different agencies across the Northern Hemisphere, are not included in NorSWE. Additional data from the Norwegian Water Resources and Energy Directorate identified through the review process that also meet our criteria will be added. Exceptionally we will also add data from the Swiss GCOS (Marty, 2020) network even though they are single point SWE measurements. This exception is being made due to the large data gap in Europe and the fact that these data are widely used and were highlighted by two reviewers. While we agree that a central repository for all snow data would be helpful but is beyond the scope and intent of our dataset.

*Marty, Christoph (2020). GCOS SWE data from 11 stations in Switzerland. EnviDat.doi:10.16904/15.*

1) Maybe not in this paper, I suggest that the authors consider including data from the California Department of Water Resources (CADWR) snow data and ASO Lidar SWE data when they update their data repository in the future.

Thank you for your suggestions of additional data.

ASO Lidar SWE: As outlined above, we purposely limit our dataset to direct estimates of SWE, although we acknowledge soil moisture is needed for the airborne gamma SWE estimates. For this reason, the ASO Lidar data are excluded because these are snow depth measurements with a model employed to go from snow depth to SWE. For the same reason we excluded SWE values from aerial markers from the NRCS network that measure snow depth and apply an assumed or modeled snow density to estimate SWE. Spatially continuous derived or interpolated datasets are incongruous with the aim of NorSWE to compile in situ SWE observations.

California Department of Water Resources (CADWR): The NRCS data we include compiles data from state agencies across the western US, including California. Indeed, the California Data Exchange Center – California Department of Water Resources (CADWR) is an excellent site and

data repository that provides accessible snow information to a range of users. Most of these data are fed into the NRCS system that we pull from.

In response to your suggestion, we compared snow course sites from CADWR with those from NorSWE within California and adjoining states (Nevada and Oregon specifically). After adjusting for differences in station names, all but two active snow course sites in CADWR are in NorSWE: Sache Springs 2 which had a route redesign in 2017 and Little Whitney Meadow. Given these similarities and the fact that we aim to reduce site duplication within NorSWE we elected to keep the NRCS data '*as-is*'. Instead, we propose the following addition to highlight the existence of state-level data portals and to clarify from which database the records in NorSWE were obtained.

L186: *"Snow survey data from the US Natural Resources Conservation Service covering the US were obtained directly using the GitHub repository https://github.com/CH-Earth/snowcourse. NRCS data compiles observations from state-level data collection offices across the western US and Alaska. Some states such as California also provide these observations through their own data portals. To ensure broad consistency across the region and to avoid introducing duplicate records we chose to draw only from the compiled NRCS database."*

2) Different data sources have distinct measurement protocols, which can make uniform quality control difficult. It may be helpful for the authors to discuss the limitations of different datasets, such as the temperature bias and the precipitation underestimation issue in SNOTEL, and reference other works that have attempted to address these challenges.

Temperature and precipitation are not included in our dataset so we did not include discussion of temperature and precipitation biases in SNOTEL. We propose to modify the following text to clearly highlight that the datasets cited aim to address biases between SWE and accumulated precipitation often attributed to drifting snow.

L124: *"Compiled and quality-controlled snow pillow data over western North America that aim to address biases between accumulated precipitation and SWE (Meyer et al. 2012) are available elsewhere (e.g. Yan et al., 2018; Sun et al., 2019; Musselman, 2021)."*

Further, we propose to add additional information about the limitations and uncertainties of the measurement techniques and of our QC procedure to address biases attributed to measurement method as follows:

L113: *"The stated accuracy of common automated passive gamma instruments is ± 15 mm up to 300 mm and 15% from 300 mm to 600 mm (Campbell Scientific, 2017) but can be as low as 5% with careful site-specific calibration (see Royer et al. and references therein). When deployed in the field, measurement uncertainty varies according to environmental factors and soil moisture conditions and may exceed the manufacturer specifications (Smith et al., 2017; Royer et al. 2021; and references therein) ... Snow pillows are prone to errors (both over and underestimates) when the temperature at the ground-snow interface is at the melting point (Johnson and Marks, 2004)."*

L222: *"To ensure reproducibility of the quality control, we chose not to implement procedures that rely on ancillary data such as precipitation and temperature (e.g. Johnson and Marks, 2004b; Yan et al., 2018; Brown et al., 2021) and instead apply only self-contained methods. Ancillary data are not always consistently available and can be subject to version changes and updates. We encourage*

*users to conduct additional QC using locally available ancillary data when possible. **Measurement uncertainty differs according to sensors type, measurement equipment and operational protocols (Sect. 2; Lopez-Moreno et al. 2020; Royer et al. 2021; Beaudoin-Galaise, and Jutras, 2022 and references therein); these differences are addressed in our QC procedure.***"

Added references

*Beaudoin-Galaise, M. and Jutras, S.: Comparison of manual snow water equivalent (SWE) measurements: seeking the reference for a true SWE value in a boreal biome, The Cryosphere, 16, 3199–3214, https://doi.org/10.5194/tc-16-3199-2022 , 2022.*

*Meyer, J. D. D., Jin, J.,and Wang, S.-Y.: Systematic Patterns of the Inconsistency between Snow Water Equivalent and Accumulated Precipitation as Reported by the Snowpack Telemetry Network, J. Hydrometeorol., 13(6), 1970–76, https://doi.org/10.1175/JHM-D-12-066.1, 2012.*

*Royer, A., Roy, A., Jutras, S., and Langlois, A.: Review Article: Performance Assessment of Radiation-Based Field Sensors for Monitoring the Water Equivalent of Snow Cover (SWE), The Cryosphere, 15(11), 5079–98, https://doi.org/10.5194/tc-15-5079-2021, 2021.*

Further, in response to Reviewer 1 (Adrià Fontrodona-Bach) we propose expanded text regarding airborne gamma SWE uncertainties as follows:

Expanded text:

L102: *"Error simulations and comparisons with coincident ground-based observations have reported accuracies of 4% to 10% in prairie and agricultural environments (Carroll et al. 1983) and up to ~12% in forested areas (Carroll and Vose, 1984; Vogel, 1985; Carroll and Carroll 1989a), although some studies have reported larger errors Glynn 1988; Cho et al. 2020a Figure 9). A comprehensive accuracy assessment of NOHRSC airborne gamma SWE showed strong correlation with the University of Arizona SWE product across all land covers and forest fractions (Cho et al. 2020b). Underestimation often occurs when there is significant SWE variability along a flight line (Cork and Loijens, 1980; Carroll and Carroll 1989b). Inaccurate characterization of the soil moisture, often due to changes in the soil moisture after the fall reference flight, is a common source of error (Carroll and Carroll 1989b; Cho et al. 2020a). Other known sources of error include biomass, rock outcrops, navigation, and gamma count statistics (Glynn et al. 1988; Cork and Carroll and Carroll 1989a; Carroll and Carroll 1989b)."*

*Added references*

*Carroll, T. R., Glynn, J.E., and Goodison, B.E.: A comparison of U.S. and Canadian airborne gamma radiation snow water equivalent measurements, Proc. West. Snow Conf., 51, 27-37, 1983.*

*Carroll, S.S., Carroll, T.R.: Effect of uneven snow cover on airborne snow water equivalent estimates obtained by measuring terrestrial gamma radiation, Water Resour. Res., 25 (7), 1505‑1510, https://doi.org/10.1029/WR025i007p01505, 1989b.*

*Carroll, S.S. and Carroll, T.R.: Effect of forest biomass on airborne snow water equivalent estimates obtained by measuring terrestrial gamma radiation. Remote Sens. Environ. 27 (3), 313‑319. https://doi.org/10.1016/0034-4257(89)90091-6, 1989a.*

*Cho, E., Jacobs, J.M., Schroeder, R., Tuttle, S. E. and Olheiser, C.: Improvement of operational airborne gamma radiation using SMAP soil moisture, Remote Sens. Environ., 240, 111668, https://doi.org/10.1016/j.rse.2020.111668, 2020a.*

*Cho, E., Jacobs, J.M., and Vuyovich, C.: The value of long-term (40 years) airborne gamma radiation SWE record for evaluating three observation-based gridded SWE datasets by seasonal snow and land cover classifications, Water Resour. Res., 56, e2019WR025813, https://doi.org/10.1029/2019WR025813, 2020b.*

*Cork, H. F. and H. S. Loijens, H.S.: The effect of snow drifting on gamma survey results, J. Hydrol., 48(1-2), 41-51, https://doi.org/10.1016/0022-1694(80)90064-5, 1980.*

*Glynn, J.E., Carroll, T.R., Holman, P.B., Grasty, R.L.: An airborne gamma ray snow survey of a forested covered area with a deep snowpack, Remote Sens. Environ., 26 (2), 149‑160, https://doi.org/10.1016/0034-4257(88)90093-4, 1980.*

*Vogel, R. M., Carroll, T. R., and Carroll, S. S.: Simulation of airborne snow water equivalent measurement errors made over a forest environment, Proceedings of the American Society of Civil Engineers Symposium, Denver, CO, p. 9, 1985.*

3) Other snow-related data sources, such as UA-SWE data for 4-km resolution over CONUS and satellite snow cover data, could also be discussed in the paper.

As outlined above, the dataset's focus is in-situ SWE observations so UA-SWE and other CONUS-specific satellite snow cover data are beyond the scope of the dataset. Datasets such as NorSWE can help inform these spatially and temporally continuous datasets. We propose adding text to Section 7 in this regard.

L349: *"Further, in situ SWE observations can help inform high quality spatially and temporally continuous gridded SWE estimates, exemplified by the University of Arizona SWE product (Broxton et al. 2019). SNOTEL SWE observations and SD from the US Cooperative network are assimilated with gridded temperature and precipitation (PRISM) data to provide high-quality daily SWE and SD estimates over the Coterminous US at a 4km spatial resolution (Zeng et al. 2018). Data contained in NorSWE could help facilitate the creation of similar datasets at both the regional and hemispheric scale."*

Added references

Broxton, P., Zeng, X. and Dawson, N.: Daily 4 km Gridded SWE and Snow Depth from Assimilated In-Situ and Modeled Data over the Conterminous US. (NSIDC-0719, Version 1). [Data Set]. Boulder, Colorado USA. NASA National Snow and Ice Data Center Distributed Active Archive Center. https://doi.org/10.5067/0GGPB220EX6A, 2019.

Zeng, X., Broxton, P., Dawson, N.: Snowpack change from 1982 to 2016 over Conterminous United States, Geophys. Res. Lett., 45,12,940-12,947, https://doi.org/10.1029/2018GL079621, 2018.

---

## Author Response (AR1)

**essd-2024-602 Author response to referee comment 1**

Original comments are in black, our responses are in blue, proposed additions and modifications in red, original manuscript text in grey. Line numbers refer to the original manuscript.

**Reviewer 1 Adrià Fontrodona-Bach**

This paper presents a dataset of manual and automated in situ measurements of snow water equivalent (SWE) over the Northern Hemisphere, which is called NorSWE. To the best of my knowledge, there is no in situ global SWE dataset publicly available to date. Currently, researchers need to compile such a dataset from each individual source every time they require it for their applications, and apply their own filters and quality checks, which is time and labour intensive. The scientific community will therefore highly benefit from this dataset, and it fits very well within the scope of the journal. The dataset is excellent, offers a wide range of applications (as the authors very well describe and demonstrate in the paper) and is especially timely as many global products and applications rely on actual SWE measurements and are being increasingly used by the community. The authors did an impressive data compilation and data curation work. The paper is also well written and clear and I hope to see it published soon. However, I have a few minor comments/suggestions and technical corrections that should be addressed before the paper is published.

I only have one rather **major suggestion**, but I call it major just because it may require a bit more time than the rest of minor comments. It regards the spatial coverage of the dataset. The dataset covers a large part of the Northern Hemisphere, but there are some gaps, which the authors recognise in Section 8 (Lines 373-380). It is true that there is a lack of observations in certain areas (e.g. high mountain Asia), and that many other SWE data are just not publicly available. I also acknowledge that it is not possible to find and include every single available dataset and that a line must be drawn somewhere. However, there is some data available over Europe which the authors did not include and which I strongly encourage they do. These include the Global Climate Observing System (GCOS), the Norwegian Water Resources and Energy Directorate, and a few other individual sites over the Alps. The sources are well listed in Table 1 in Fontrodona-Bach et al. (2023), and they are also used and listed by Seo et al. (2025) in Table 1 https://doi.org/10.5194/essd-2024-349 (preprint). This would give some coverage (despite being still limited) to the European Alps and to Scandinavia outside of Finland, and I strongly encourage that these datasets are included. If the authors wish, they can get in touch with me and I will send them some notes on how I downloaded these data.

Thank you for the constructive feedback.

We added data from the Norwegian Water Resources and Energy Directorate (NVE) and the Swiss Federal Research Institute (WSL) – Institute for Snow and Avalanche Research (SLF): *'GCOS SWE data from 11 stations in Switzerland'* (Marty 2020). The NVE data includes six snow courses and 19 automated sites, one of which is located in Nepal. All tables, figures (see end of this document) and relevant text has been updated accordingly. The Norway (NVE) data rea detailed in Section 3.5 (L254-265).  The Swiss data do not fit our criteria of snow courses or airborne measurements for SWE but are exceptionally included to address the data gap in Europe.  The Single point manual measurements – Switzerland are detailed in Section 2.1 (L109-114).

*Marty, Christoph (2020). GCOS SWE data from 11 stations in Switzerland. EnviDat.doi:10.16904/15.*

I do not know if the authors requested permission to each individual agency to include their data in NorSWE (if they did, maybe they should explain this in the paper). In any case, in my opinion it is necessary to include clear statements that when using NorSWE data, all the original data sources (so all source datasets) must be appropriately cited as well as the citation of this paper and the NorSWE dataset itself (which is on Zenodo). This provides clear and proper acknowledgement to previous data collections that form this compilations dataset.

We made every attempt to identify the licensing and data redistribution policies associated with each dataset either through direct email with data providers or by investigating the corresponding websites. Indeed, this is the main reason this manuscript and dataset took a long time to come to fruition. Data permissions and redistribution information is included in the NetCDF general attribute 'Distribution'. However, we agree with you that this information is not clearly articulated in our text. Our intent with this dataset is certainly not to take ownership or acknowledgement away from the original data sources. We added the following acknowledgements section, which should have been included originally.

**L489-501:**

*"Acknowledgements*

*We gratefully acknowledge the field observers collecting manual snow observations and maintaining automatic stations as well as the agencies and personnel who maintaining these records. The authors of NorSWE do not own any of these data. Data are redistributed under the following licences without guarantee of the quality/accuracy of these data. Data from The Norwegian Water Resources and Energy Directorate are distributed, and modified as described herein, under the Norwegian licence for Open Government data (NLOD). CanSWE data are redistributed under the Open Government Licence - Canada. (https://open.canada.ca/en/open-government-licence-canada). Hydro-Québec data (http://www.hydroquebec.com/) are available under the terms of a Creative Commons Attribution - Non Commercial - Share A Like 4.0 International (https://creativecommons.org/licenses/by-nc-sa/4.0/) (CC BY-NC-SA 4.0). Finnish Environmental Institute (SYKE) data are redistributed under Creative Commons Attribution 4.0 International (CC BY 4.0).)), 'GCOS SWE data from 11 stations in Switzerland' are redistributed under the Open Database License (ODbL) 'Attribution Share-Alike for Databases'.*

*The authors of NorSWE have modified the original data as outlined in the manuscript. We recommend users of NorSWE reference the data providers of the relevant agencies as outlined in Table 4."*

I think this dataset will be very useful and I look forward to seeing this paper published and the dataset in use. The rest of comments are listed in the attached pdf.

Thank you for your positive and constructive feedback.

**List of minor comments and technical corrections, with line number (L):**

L 8-9: Revise sentence, I suggest "Here we present the Northern Hemisphere in situ snow water equivalent dataset (NorSWE), consisting …"
Suggestion implemented.

L 13: Perhaps a little unimportant, but since you reference the guide to methods and observations from the WMO (2018) in line 25 for the definition of SWE, then I think the acronym HS should be used for snow depth instead of SD throughout the paper and in the dataset, as stated in this same manual.
We recognize that the formal WMO acronym for snow depth is HS. However, since SD is more commonly used we will keep it in our paper. Upon the first use of snow depth in the main body of the text we will add explicit mention that HS is the formal abbreviation according to WMO.
*"…situ snow depth (SD), formally abbreviated as HS (WMO, 2018),"*

L 22: I do not understand how the references "National Academies of Sciences, Engineering, and Medicine, 2018; GCOS, 2022" support the statement of "the seasonal snowpack being critical for ecosystems and climate monitoring". Please be more specific or add more relevant references.
Replaced with Meredith et al. (2019), Thornton et al. (2021), and Gottlieb and Mankin (2024).

Added references
Thornton, J. M., Palazzi, E., Pepin, N.C., Cristofanelli, P., Essery, R., Kotlarski, S., Giuliani, G., Guigoz, Y., Kulonen, A., Pritchard, D., Li, X., Fowler, H.J., Randin, C.F., Shahgedanova, M., Steinbacher, M., Zebisch M., and Adler, C.: Toward a Definition of Essential Mountain Climate Variables, *One Earth* 4(6): 805–27, https://doi.org/10.1016/j.oneear.2021.05.005, 2021.

Meredith, M., Sommerkorn, M., Cassotta, S., Derksen, C., Ekaykin, A., Hollowed, A., Kofinas, G., Mackintosh, A., J. Melbourne-Thomas, J., Muelbert, M.M.C., Ottersen, G., Pritchard, H. and Schuur, E.A.G.: Polar Regions. In: IPCC Special Report on the Ocean and Cryosphere in a Changing Climate [Pörtner, H.-O., Roberts, D.C., Masson-Delmotte, V., Zhai, P., Tignor, M., Poloczanska, E., Mintenbeck, K., Alegría, A., Nicolai, M., Okem, A., Petzold, J., Rama, B., Weyer, N.M. (eds.)], https://www.ipcc.ch/site/assets/uploads/sites/3/2019/12/SROCC_FullReport_FINAL.pdf, 2019.

L26: Missing a reference for SWE in Global Climate Models.
Added Mudryk et al. (2020) which is concerned with snow in CMIP6.

Mudryk, L., Santolaria-Otín, M., Krinner, G., Ménégoz, M., Derksen, C., Brutel-Vuilmet, C, Brady, M., and Essery, R.: Historical Northern Hemisphere snow cover trends and projected changes in the CMIP6 multi-model ensemble, The Cryosphere, 14(7), 2495–2514, https://doi.org/10.5194/tc-14-2495-2020, 2020.

Caption Table 1: Here you are calling the dataset "NH in situ SWE" instead of "NorSWE". This also occurs on Table 5 and 7, and line 232, and maybe elsewhere, please check and modify.
Thank you for catching this and we apologize for these sloppy errors. Instances of "NH in situ SWE" replaced with "NorSWE".

Table 1: On MGS entry, remove "is" from "Bulk density is derived …" for consistency with other

entries.

Again, thank you for catching this inconsistency. Change implemented.

Table 2: Footnotes (1,2,3) are missing. I guess those would explain the difference between each quality flag?

Footnotes added. We also added text to Section 3 to clarify the difference between agency and QC flags and to provide more information about the agency flags. Table added (Table 8) to describe the QC flags applied.

Revised Table 2 with footnotes:

**Table 2: Description of variables in the NorSWE NetCDF file.**

| Type of variable | Variable name | Description | Dimension | Units |
|---|---|---|---|---|
| Dimension | station_id | Station identification code | station_id | (-) |
| | time | Time | time | Day |
| Observational metadata | lat | Station latitude | station_id | °North |
| | lon | Station longitude | station_id | ° |
| | elevation | Station elevation | station_id | m |
| | source | Data provider | station_id | (-) |
| | station_name | Station name | station_id | (-) |
| | type_mes | Method of measurement for SWE[1] | station_id | (-) |
| | mmask | Mountain mask[2] | station_id | (-) |
| Data | snw | Water equivalent of snow cover (SWE) | station_id, time | $kg\,m^{-2}$ |
| | snd | Snow depth (SD) | station_id, time | m |
| | den | Bulk snow density | station_id, time | $kg\,m^{-3}$ |
| Quality control flags | data_flag_snw | Agency data quality flag for SWE[3] | station_id, time | (-) |
| | data_flag_snd | Agency data quality flag for SWE[3] | station_id, time | (-) |
| | qc_flag_snw | CanSWE quality control flag for SWE[4] | station_id, time | (-) |
| | qc_flag_snd | CanSWE quality control flag for SD[4] | station_id, time | (-) |

1: see Table 3
2: see Section 3
3: see Table 5
4: see Table 6

190-199: *"NorSWE includes two types of flags describing the data quality: agency quality flags and QC flags. QC flags indicate where an observation did not pass our internal quality control (See **Sect. 4**) and was set to nan. Agency quality flags (**Table 5**) incorporate information from the data provider. These can be flags assigned by the agency to indicate certain snow conditions, for example patchy or trace amounts of snow, or to flag observations that were modified or removed during the agency's QC procedures, for example revised data ('R'). Some agencies include one or more comments along with each observation instead of data flags. We coded these comments into flag values using keywords and phrases. For example, records with comments 'skiff' or 'patchy' were assigned a 'T' flag. An exception to the use of the agency quality flag field is the airborne gamma SWE data which did not have corresponding agency quality information. Instead, we use this variable to store information about the soil moisture estimation method (**Sect. 3.6**)."*

**Table 8: Quality control flags in NorSWE. NaN stands for not a number.**

| QC flag | Definition |
|---|---|

| | |
|---|---|
| H | SD > 3m (> 8m in mountains). SD set to *NaN*. |
| M | Data masked (set to *NaN*) in the precursor to CanSWE. |
| V | Automatic SD-SWE measurement identified as outlier using robust Mahalanobis distance. SD and SWE set to *NaN*. |
| W | SWE > 3000 kgm-2 (> 8000 kgm$^{-2}$ in mountains). SWE set to *NaN*. |
| D | Derived bulk snow density failed 25–700 kgm$^{-3}$ threshold. SD, SWE and bulk snow density set to *NaN*. |

Table 3: I think "64" should be removed from the caption.
Table 3 (L85) caption revised to: **Table 3: WMO BUFR (WMO, 2019) SWE measurement codes and non-WMO code for airborne gamma SWE.**

Table 3: What does "7-62 Reserved mean"?
Table 3, except value 64, is taken directly from the WMO SWE measurement codes (WMO, 2019). Our understanding is that numbers 7 – 62 are reserved for future use – i.e. future methods requiring specific codes. Code 3 'passive gamma' does not differentiate between automated gamma measurements from airborne measurements, but these are very different implementations of 'passive gamma' methods, so we added a 'new' (non-WMO) code for airborne gamma specifically. Because 7 – 62 are 'reserved' and 63 is intended for missing measurement type values we assigned airborne gamma the next available numeric code – 64.

L89: Not sure the abbreviation cf. is properly used here? I had not seen this before, but while searching what it means I found that it is sometimes wrongly used in science.
https://scientistseessquirrel.wordpress.com/2016/06/13/friends-dont-let-friends-use-cf/
'cf.' removed

Lines 94-105: Does this paragraph belong here? It is in section 2.1 on Manual gravimetric snow surveys but the paragraph is about flight surveys and the gamma radiation method.
Indeed, Section 2.2 subheading was missing. We added a subheading  add subheading *2.3 Airborne gamma SWE* (L146) and moved this section after the automated measurements (Section 2.2).

L 102: What are GM and GI? Should it be "or" instead of "of"?
Corrected. Should be 'or'.

L 104: Is the uncertainty of the estimates so precise (23 mm)? But I also wonder how reliable these uncertainty ranges are applied to nowadays, considering the papers cited are from 1983 and 1984.

The majority of the literature regarding airborne gamma SWE is from the 1980s. Recent work by Eunsang Cho has demonstrated good agreement with the University of Arizona SWE dataset and also included some comparisons with ground observations. We propose to expand the discussion of uncertainties related to airborne gamma SWE. We acknowledge many of the added references are still quite old.

Expanded text:

155-164: *"Error simulations and comparisons with coincident ground-based observations have reported accuracies of 4% to 10% in prairie and agricultural environments (Carroll et al., 1983) and up to ~12% in forested areas (Carroll and Vose, 1984; Vogel, 1985; Carroll and Carroll, 1989a), although some studies have reported larger errors (Glynn, 1988; Figure 9 in Cho et al., 2020a). A*

*comprehensive accuracy assessment of NOHRSC airborne gamma SWE showed strong correlation with the University of Arizona SWE product across all land covers and forest fractions (Cho et al., 2020b). Underestimation often occurs when there is significant SWE variability along a flight line (Cork and Loijens, 1980; Carroll and Carroll, 1989b). Inaccurate characterization of the soil moisture, often due to changes in the soil moisture after the fall reference flight, is a common source of error (Carroll and Carroll, 1989b; Cho et al., 2020a). Other known sources of error include biomass, rock outcrops, navigation, and gamma count statistics (Glynn et al., 1988; Cork and Loijens, 1980; Carroll and Carroll, 1989b)."*

*Added references*

*Carroll, T. R., Glynn, J.E., and Goodison, B.E.: A comparison of U.S. and Canadian airborne gamma radiation snow water equivalent measurements, Proc. West. Snow Conf., 51, 27-37, 1983.*

*Carroll, S.S., Carroll, T.R.: Effect of uneven snow cover on airborne snow water equivalent estimates obtained by measuring terrestrial gamma radiation, Water Resour. Res., 25 (7), 1505-1510, https://doi.org/10.1029/WR025i007p01505, 1989b.*

*Carroll, S.S. and Carroll, T.R.: Effect of forest biomass on airborne snow water equivalent estimates obtained by measuring terrestrial gamma radiation. Remote Sens. Environ. 27 (3), 313-319. https://doi.org/10.1016/0034-4257(89)90091-6, 1989a.*

*Cho, E., Jacobs, J.M., Schroeder, R., Tuttle, S. E. and Olheiser, C.: Improvement of operational airborne gamma radiation using SMAP soil moisture, Remote Sens. Environ., 240, 111668, https://doi.org/10.1016/j.rse.2020.111668, 2020a.*

*Cho, E., Jacobs, J.M., and Vuyovich, C.: The value of long-term (40 years) airborne gamma radiation SWE record for evaluating three observation-based gridded SWE datasets by seasonal snow and land cover classifications, Water Resour. Res., 56, e2019WR025813, https://doi.org/10.1029/2019WR025813, 2020b.*

*Cork, H. F. and H. S. Loijens, H.S.: The effect of snow drifting on gamma survey results, J. Hydrol., 48(1-2), 41-51, https://doi.org/10.1016/0022-1694(80)90064-5, 1980.*

*Glynn, J.E., Carroll, T.R., Holman, P.B., Grasty, R.L.: An airborne gamma ray snow survey of a forested covered area with a deep snowpack, Remote Sens. Environ., 26 (2), 149-160, https://doi.org/10.1016/0034-4257(88)90093-4, 1980.*

*Vogel, R. M., Carroll, T. R., and Carroll, S. S.: Simulation of airborne snow water equivalent measurement errors made over a forest environment, Proceedings of the American Society of Civil Engineers Symposium, Denver, CO, p. 9, 1985.*

L 114: "can **be** much larger"
Correction made.

Figure 1: add "a,b,c" panels, and perhaps add "Surveys" or "Manual" in panel a?
Will added 'a', 'b', 'c' added to panels and renamed 'a' – manual.

L 129: "Data from each of source".

Corrected to 'Data from each source listed ...'

L 136: What does "Harmonizing agency-specific quality flags" mean? I got a little confused, because as I understand from Table 2, agency specific quality flags are kept as in the original format in the dataset, in the field "data_flag_snw" and "data_flag_snd", but then in Table 5 the flags are "harmonized", so it is not clear if flags are modified or used as they are. Furthermore, you provide additional quality control flags (Section 4), but these are not specified in Table 2 or 5 (e.g. quality flag "H" in line 235 or "D" in line 236), unless by "CanSWE quality control flag" in Table 2 you mean "NorSWE quality control flag"?. Is there a table or variable missing with your own quality control flags? And how are these distinguished from the original flags from the agencies?

Thank you for this comment. It is clear we did not adequately explain the difference between agency and QC flags, and our text discussing 'agency quality flags' is insufficient.
We added the following text to Section 3 to better provide a better explanation of 'agency quality flags' and to offer better distinction between them and QC flags. We will also add a table describing the QC flags (new Table 8).

190-199: *"NorSWE includes two types of flags describing the data quality: agency quality flags and QC flags. QC flags indicate where an observation did not pass our internal quality control (See **Sect. 4**) and was set to nan. Agency quality flags (**Table 5**) incorporate information from the data provider. These can be flags assigned by the agency to indicate certain snow conditions, for example patchy or trace amounts of snow, or to flag observations that were modified or removed during the agency's QC procedures, for example revised data ('R'). Some agencies include one or more comments along with each observation instead of data flags. We coded these comments into flag values using keywords and phrases. For example, records with comments 'skiff' or 'patchy' were assigned a 'T' flag. An exception to the use of the agency quality flag field is the airborne gamma SWE data which did not have corresponding agency quality information. Instead, we use this variable to store information about the soil moisture estimation method (**Sect. 3.6**)."*

**Table 8: Quality control flags in NorSWE. NaN stands for not a number.**

| QC flag | Definition |
| --- | --- |
| H | SD > 3m (> 8m in mountains). SD set to *NaN*. |
| M | Data masked (set to *NaN*) in the precursor to CanSWE. |
| V | Automatic SD-SWE measurement identified as outlier using robust Mahalanobis distance. SD and SWE set to *NaN*. |
| W | SWE > 3000 kgm-2 (> 8000 kgm$^{-2}$ in mountains). SWE set to *NaN*. |
| D | Derived bulk snow density failed 25–700 kgm$^{-3}$ threshold. SD, SWE and bulk snow density set to *NaN*. |

L139: What does "using unique agency-specific Python scripts" mean? Please specify.
'using unique agency-specific Python scripts' removed.

L139: coordinate**s**
Corrected

Table 5: What does "revised data" mean? And what is the difference between Traces and Patches?

Thank you for the question regarding 'Trace' versus 'Patches'. We consolidated these into a single 'Trace' flag. Added to L195-196: For example, records with comments 'skiff' or 'patchy' were assigned a 'T' flag. Table 5 was updated accordingly.

Revised data means the data were revised by the original providing agency. This is now clarified in the expanded Section 3 text (above), relevant excerpt:
*"...to flag observations that were modified or removed during their internal QC procedures, for example revised data."*

**Table 5: Data flags in NorSWE. Not all data flags are used by all data sources.**

| Data flag | Definition | Comment |
|---|---|---|
| A | Sampling problems | |
| B | Manual snow survey conducted outside the nominal sampling period | |
| C | Combination of A and B | |
| E | Estimate | |
| G | Measurement location > 1 km from station coordinate. | Specific to Saskatchewan Water Security Agency housed in CanSWE beginning in 2011. |
| M | Missing | |
| R | Revised data | |
| T | Trace | Includes patchy snow conditions |
| Y | Precise sampling date not available. CanSWE: NWT set to 1 April, within 1 week for Government of Manitoba, research sites (UU) approximate date. NRCS: set to nominal survey date. | |
| AI | Soil moisture – airborne | Data flags for airborne gamma are |
| AM | Soil moisture – airborne | used to store the soil moisture |
| GI | Soil moisture – ground-based information | estimation method. |
| SE | Soil moisture - subjective estimate | |
| MM-avgX | Average of 'X' SWE observations using soil moisture method 'MM' where 'MM' is the soil moisture method AI, AM, GI or SE. | Specific to airborne gamma SWE Lines with >1 observation on a given date, see Section 3.6. |

Section 3: I find the titles of the subsections a little inconsistent, as some subtitles refer to specific datasets (3.1, 3.2, 3.4), one to a region (3.3) and another one to a method (3.5). I suggest to harmonise them by naming Subsection 3.3 as "Northeast US (MGS, NRCC, NHDES)" and Subsection 3.5 as "Airborne gamma SWE (NOHRSC)", so that at least all datasets are mentioned. Or any other harmonisation that makes the subsectioning clearer.

Agreed. Section 3 subheadings have been revised as follows:

3.1 Canada (CanSWE)
3.2 Russia (RIHMI-WDC)
3.3 Northeast US (Maine GS, NH-DES, NRCC)
3.4 Western US and Alaska (NRCS)
3.5 Norway (NVE)

3.6 US and southern Canada airborne gamma SWE (NOHRSC)

L148-149: Please explain why SYKE (Finland) did not require any additional processing steps.
The data from SYKE was clean and did not include any agency data flags so extra processing steps were required. Only the general processing steps were applied. This was also the case for the Switzerland data that was added.

L169: Why is the land cover type not supported by NetCDF? Please explain.
Text in question removed.

Table 6: "Station coordinates  **are** the same ..." and "RIHMI instead of RHIMI.
Corrected

L 234: I think this should be Table 5 and not 7?
Revised to refer to new Table 8 (other Tables renumbered accordingly).

**Table 8: Quality control flags in NorSWE. NaN stands for not a number.**

| QC flag | Definition |
|---------|-----------|
| H | SD > 3m (> 8m in mountains). SD set to *NaN*. |
| M | Data masked (set to *NaN*) in the precursor to CanSWE. |
| V | Automatic SD-SWE measurement identified as outlier using robust Mahalanobis distance. SD and SWE set to *NaN*. |
| W | SWE > 3000 kgm-2 (> 8000 kgm$^{-2}$ in mountains). SWE set to *NaN*. |
| D | Derived bulk snow density failed 25–700 kgm$^{-3}$ threshold. SD, SWE and bulk snow density set to *NaN*. |

L 243: qg_fladg_snd
Corrected

L 250 and 253: Please specify what "similar coordinates" means.
Text revised for clarity.
L325-328: *"Duplicate sites were defined as those with similar locations, snow observations, station names or IDs as follows. First, we identified all sites from neighbouring agencies with matching station names and inspected those matched sites within 5 km of each other. If the paired sites had matching coordinates and snow records (within rounding precision), we retained the site from the agency whose jurisdiction it intersects."*

L 259: "This step removed 63 sites: 62 + 7"? Shouldn't it be 69?
According to Table S1 it should read *"This step removed 63 sites: 56 from CanSWE and 7 from Maine GS (**Table S1**)."*

L 264: I suggest to be specific with the exact number of observations in this section, and the exact number of sites.
As per your suggestion we will provide exact numbers for the revised dataset.
We have provided exact numbers in the Dataset summary (Setc. 6), L337-345:
*"The final dataset contains 11,593,790 valid SWE observations (11,762,366 when including flagged values set to nan) from 10,153 different sites. There are 947,962 SWE observations from 6,672 snow course sites across North America, Norway, Finland and Russia and an additional 4,000*

*observations from eleven single point snow pit sites in Switzerland. Together, 2,357 airborne gamma flight lines provide 30,889 SWE observations over the US and southern Canada. Automated sites account for only 11% of the sites in NorSWE but owing to their higher sampling frequency account for 92% of the SWE observations. The 1,113 automated sites are largely restricted to North America, with 17 in Norway and one in Nepal. NorSWE includes 10,700,949 snow pillow observations from 1001 sites across western North America, Norway and Nepal and 134,216 GMON observations from 113 sites in Québec and Newfoundland and Labrador, Canada, and Norway."*

L 273: I think it should be Fig. 3 or Fig. 5 but not Fig. 4?
Corrected to Fig. 3.

Figure 3: I suggest to rephrase the caption, the first three lines are long and without a comma.
Figure 3 caption revised:

**Figure 3. (a) NorSWE site distribution by snow class (Sturm and Liston, 2021) for the complete dataset (solid bars) and a temporally consistent subset (hatched bars) versus the proportional land area by snow class (dashed black line). The temporally consistent subset consists of sites with at least one measurement in each pentad starting in 1980 and having measurements in at least 30 different years between 1979 and 2021. The ephemeral snow class is excluded from the land area calculations because it does not differentiate between no snow and ephemeral. Permanent land ice is also excluded. (b) Map showing the geographical extent of Sturm and Liston (2021) snow classes. Montane: montane forest, Boreal: boreal forest.**

L 338: I suggest to remove "briefly", as it is not very brief. Great section though! It is very strong describing so many data usages, well done.
Suggestion implemented.

L 344: "uses **to** parameterize"
Corrected.

L 355-357: Yes absolutely! That is why this dataset is great, when I did the data collection myself, I missed these eastern US datasets. Great job!
Your dataset is great. We just thought the differences in North America were quite interested. As you know, data collation is challenging and takes a village, so we greatly appreciate your suggestions about additional European data.

L 379: Not sure how, but it would be great to include a reference/link for this https://mountainresearchinitiative.org/flagship-activities/joint-body-on-the-status-of-mountain-snow-cover/
Agreed. We added a link to this activity:
"Recent initiatives such as the WMO Joint Body for the Status of Mountain Snow Cover (https://mountainresearchinitiative.org/flagship-activities/joint-body-on-the-status-of-mountain-snow-cover/) may uncover additional sources that could be included in a future version of NorSWE."

Table A1: Perhaps specify for each entry if it is CanSWE or NorSWE that they used? Assuming some of those already used NorSWE. It was not entirely clear to me if the table is about showing examples of "potential uses", or studies that have already used these specific datasets.

Table A1 is intended to show the ranges of uses of this type of dataset. Title renamed accordingly and dataset used added.

Table A1. Demonstration of uses of NorSWE and its precursor CanSWE. NorSWE includes v1 and its unpublished precursors. Uses of Snotel data are detailed in Flemming et al. (2023).

| Benchmarking gridded SWE products | | Dataset |
|---|---|---|
| **Mortimer et al. 2022**
https://doi.org/10.1016/j.rse.2022.112988 | Benchmarking EO SWE product (Snow CCI+) | NorSWE |
| **Luojus et al. 2021**
https://doi.org/10.1038/s41597-021-00939-2 | Validation of GlobSnow v3 product and older CHSSD dataset (Brown et al., 2019) used as input to monthly bias correction. | NorSWE |
| **Gao et al. 2023**
https://doi.org/10.3390/rs15082065 | Evaluation of snow densities derived from SMOS over Quebec, Canada. | CanSWE |
| **Mortimer et al. 2024**
https://doi.org/10.5194/tc-18-5619-2024 | Impact of in situ method on benchmarking gridded SWE products. | NorSWE |
| **Mudryk et al. 2025**     https://doi.org/10.5194/tc-18-5619-2024 | Benchmarking of 23 gridded products from the SnowPex+ intercomparison project. | NorSWE |
| **Elias Chereque et al. 2024**
https://doi.org/10.5194/tc-18-4955-2024 | Evaluation of simple temperature index model with different meteorological forcings. | NorSWE |
| **Sun et al. 2024**
https://doi.org/10.5194/egusphere-2024-3213 | Evaluation of a mountain SWE reanalysis with snow cover fraction data assimilation. | NorSWE |

| Hydrological model development and evaluation | | |
|---|---|---|
| **Garnaud et al. 2021**
https://doi.org/10.3390/rs13245022 | Evaluation of snow analyses in hydrological models for forecasting. | CanSWE |
| **Arnal et al. 2024**
https://doi.org/10.5194/hess-28-4127-2024 | Seasonal hydrological forecasting. | NorSWE |
| **Mai et al. 2022**
https://doi.org/10.5194/hess-26-3537-2022 | Evaluation and selection of reference datasets for Great Lakes Runoff Intercomparison Project. | CanSWE |
| **Marsh et al. 2024**
https://doi.org/10.1029/2023WR036948 | Evaluation of simulated snow drifting patterns with the Canadian Hydrological Model across the Canadian cordillera and adjacent regions. | CanSWE |
| **Shrestha et al. 2022**
https://doi.org/10.3389/frwa.2022.801134 | Evaluation of a functional hydrological model of the Great Lakes Basin. | CanSWE |
| **Vionnet et al. 2022**
https://doi.org/10.1029/2021WR031778 | Evaluation of the ability of precipitation phase information to improve mountain snowpack prediction. | CanSWE |

| Model input, parameterization, retrieval schemes | | |
|---|---|---|
| **Fontrodona-Bach et al. 2023**
https://doi.org/10.5194/essd-15-2577-2023 | CanSWE (and Snotel) data used to develop snow density model to go from SD to SWE. (NH-SWE). | CanSWE |
| **Venäläinen et al. 2023**
https://doi.org/10.5194/tc-17-719-2023 | Interpolated in situ snow density information for use in GlobSnow SWE retrieval. | NorSWE |
| **Dulfer et al. 2022**
https://doi.org/10.1016/j.quascirev.2022.107465 | CanSWE (SD and density) used to calculate snow shielding factors. | CanSWE |
| **Tian et al. 2024**
https://doi.org/10.1016/j.scs.2024.105660 | Training data for machine learning model to investigate the reliability of rapid public transit in the Toronto region under various climate change scenarios. | CanSWE |

| Snow status and trends | | |
|---|---|---|

| | | |
|---|---|---|
| **Gottlieb and Mankin 2024**
https://doi.org/10.1038/s41586-023-06794-y | Observational data (CanSWE) used to in attribution study of impact of human influence on NH snow loss. | CanSWE |
| **Hale et al. 2023**
https://doi.org/10.1038/s43247-023-00751-3 | Changes in snow water storage. CanSWE data from 1 April used to evaluate snow storage index output from Snow Storage Index. | CanSWE |
| | **Other** | |
| **Pokorny et al 2023**
https://doi.org/10.1061/JHYEFF.HEENG-5833 | Uncertainty analysis – model uncertainties | CanSWE |

**essd-2024-602 Author response to referee comment 2**

Original comments are in black, our responses are in blue, proposed additions and modifications in red, original manuscript text in grey. Line numbers refer to the original manuscript.

**Reviewer 2**

I appreciate the authors' effort in creating a centralized data repository for all snow data. I have a few suggestions to further enhance the paper:

The purpose of the dataset is not a centralized repository for all snow data but rather a compilation of in situ SWE measurements. The dataset was initially compiled to support evaluation of gridded SWE products so was limited to snow course and airborne gamma SWE which are more spatially representative than single point measurements. It was later expanded to include automated SWE measurements covering North America to support evaluation of hydrological models. These criteria exclude derived data and those based on satellite observations. Further we stress that the dataset's focus is on SWE, not snow cover. Snow depth and snow density are only included when available alongside the SWE observations. Consequently, snow depth measurements from automatic stations, non collocated to SWE measurements, which are collected by many different agencies across the Northern Hemisphere, are not included in NorSWE. While we agree that a central repository for all snow data would be helpful but is beyond the scope and intent of our dataset.

Following feedback from reviewers we added data from the Norwegian Water Resources and Energy Directorate (NVE) and the Swiss Federal Research Institute (WSL) – Institute for Snow and Avalanche Research (SLF): *'GCOS SWE data from 11 stations in Switzerland'* (Marty 2020). The NVE data includes six snow courses and 19 automated sites, one of which is located in Nepal. All tables, figures and relevant text has been updated accordingly. The Norway (NVE) data rea detailed in Section 3.5 (L254-265). The Swiss data do not fit our criteria of snow courses or airborne measurements for SWE but are exceptionally included to address the data gap in Europe. The Single point manual measurements – Switzerland are detailed in Section 2.1 (L109-114).

*Marty, Christoph (2020). GCOS SWE data from 11 stations in Switzerland. EnviDat.doi:10.16904/15.*

1) Maybe not in this paper, I suggest that the authors consider including data from the California Department of Water Resources (CADWR) snow data and ASO Lidar SWE data when they update their data repository in the future.

Thank you for your suggestions of additional data.

ASO Lidar SWE: As outlined above, we purposely limit our dataset to direct estimates of SWE, although we acknowledge soil moisture is needed for the airborne gamma SWE estimates. For this reason, the ASO Lidar data are excluded because these are snow depth measurements with a model employed to go from snow depth to SWE. For the same reason we excluded SWE values from aerial markers from the NRCS network that measure snow depth and apply an assumed or modeled snow density to estimate SWE. Spatially continuous derived or interpolated datasets are incongruous with the aim of NorSWE to compile in situ SWE observations.

California Department of Water Resources (CADWR): The NRCS data we include compiles data from state agencies across the western US, including California. Indeed, the California Data Exchange Center – California Department of Water Resources (CADWR) is an excellent site and data repository that provides accessible snow information to a range of users. Most of these data are fed into the NRCS system that we pull from.

In response to your suggestion, we compared snow course sites from CADWR with those from NorSWE within California and adjoining states (Nevada and Oregon specifically). After adjusting for differences in station names, all but two active snow course sites in CADWR are in NorSWE: Sache Springs 2 which had a route redesign in 2017 and Little Whitney Meadow. Given these similarities and the fact that we aim to reduce site duplication within NorSWE we elected to keep the NRCS data '*as-is*'. Instead, we propose the following addition to highlight the existence of state-level data portals and to clarify from which database the records in NorSWE were obtained.

L241-245: *"Snow survey data from the US Natural Resources Conservation Service were obtained directly using the GitHub repository https://github.com/CH-Earth/snowcourse. NRCS data compiles observations from state-level data collection offices across the western US and Alaska. Some states such as California (https://cdec.water.ca.gov/snow.html) also provide these observations through their own data portals. To ensure broad consistency across the region and to avoid introducing duplicate records we chose to draw only from the NRCS database."*

2) Different data sources have distinct measurement protocols, which can make uniform quality control difficult. It may be helpful for the authors to discuss the limitations of different datasets, such as the temperature bias and the precipitation underestimation issue in SNOTEL, and reference other works that have attempted to address these challenges.

Temperature and precipitation are not included in our dataset so we did not include discussion of temperature and precipitation biases in SNOTEL. We modified the text to clearly highlight that the datasets cited aim to address biases between SWE and accumulated precipitation often attributed to drifting snow.

L139-141: *"Compiled and quality-controlled snow pillow datasets for western North America that aim to address biases between accumulated precipitation and SWE (Meyer et al., 2021) are available elsewhere (e.g. Yan et al., 2018; Sun et al., 2019; Musselman, 2021)."*

Further, we added information about the limitations and uncertainties of the measurement techniques and of our QC procedure to address biases attributed to measurement method as follows:

L125-127: *"The stated accuracy of common automated passive gamma instruments is ± 15 mm up to 300 mm and 15% from 300 mm to 600 mm (Campbell Scientific, 2017) but can be as low as 5% with careful site-specific calibration (see Royer et al., 2021 and references therein). When deployed in the field, measurement uncertainty varies according to environmental factors and soil moisture conditions and may exceed the manufacturer specifications (Stranden et al., 2015; Smith et al., 2017; Royer et al., 2021)."*

L137-138: *"Snow pillows are prone to errors (both over and underestimates) when the temperature at the ground-snow interface is at the melting point (Johnson and Marks, 2004)."*

L290-296: *"To ensure reproducibility of the quality control, we chose not to implement procedures that rely on ancillary data such as precipitation and temperature (e.g. Johnson and Marks, 2004; Yan et al., 2018; Brown et al., 2021) and instead apply only self-contained methods. Ancillary data are not always consistently available and can be subject to version changes and updates. We encourage users to conduct additional QC using locally available ancillary data when possible. Measurement uncertainty differs according to sensors type, measurement equipment and operational protocols (Sect. 2; López-Moreno et al. 2020; Royer et al., 2021; Beaudoin-Galaise and Jutras, 2022 and references therein); these differences are not addressed in our QC procedure."*

Added references

*Beaudoin-Galaise, M. and Jutras, S.: Comparison of manual snow water equivalent (SWE) measurements: seeking the reference for a true SWE value in a boreal biome, The Cryosphere, 16, 3199–3214, https://doi.org/10.5194/tc-16-3199-2022 , 2022.*

*Meyer, J. D. D., Jin, J.,and Wang, S.-Y.: Systematic Patterns of the Inconsistency between Snow Water Equivalent and Accumulated Precipitation as Reported by the Snowpack Telemetry Network, J. Hydrometeorol., 13(6), 1970–76, https://doi.org/10.1175/JHM-D-12-066.1, 2012.*

*Royer, A., Roy, A., Jutras, S., and Langlois, A.: Review Article: Performance Assessment of Radiation-Based Field Sensors for Monitoring the Water Equivalent of Snow Cover (SWE), The Cryosphere, 15(11), 5079–98, https://doi.org/10.5194/tc-15-5079-2021, 2021.*

Further, in response to Reviewer 1 (Adrià Fontrodona-Bach) we added text regarding airborne gamma SWE uncertainties as follows:

Expanded text:

155-164: *"Error simulations and comparisons with coincident ground-based observations have reported accuracies of 4% to 10% in prairie and agricultural environments (Carroll et al., 1983) and up to ~12% in forested areas (Carroll and Vose, 1984; Vogel, 1985; Carroll and Carroll, 1989a), although some studies have reported larger errors (Glynn, 1988; Figure 9 in Cho et al., 2020a). A comprehensive accuracy assessment of NOHRSC airborne gamma SWE showed strong correlation with the University of Arizona SWE product across all land covers and forest fractions (Cho et al., 2020b). Underestimation often occurs when there is significant SWE variability along a flight line (Cork and Loijens, 1980; Carroll and Carroll, 1989b). Inaccurate characterization of the soil moisture, often due to changes in the soil moisture after the fall reference flight, is a common source of error (Carroll and Carroll, 1989b; Cho et al., 2020a). Other known sources of error include biomass, rock outcrops, navigation, and gamma count statistics (Glynn et al., 1988; Cork and Loijens, 1980; Carroll and Carroll, 1989b)."*

*Added references*

*Carroll, T. R., Glynn, J.E., and Goodison, B.E.: A comparison of U.S. and Canadian airborne gamma radiation snow water equivalent measurements, Proc. West. Snow Conf., 51, 27-37, 1983.*

*Carroll, S.S., Carroll, T.R.: Effect of uneven snow cover on airborne snow water equivalent estimates obtained by measuring terrestrial gamma radiation, Water Resour. Res., 25 (7), 1505‑1510, https://doi.org/10.1029/WR025i007p01505, 1989b.*

*Carroll, S.S. and Carroll, T.R.: Effect of forest biomass on airborne snow water equivalent estimates obtained by measuring terrestrial gamma radiation. Remote Sens. Environ. 27 (3), 313 -319. https://doi.org/10.1016/0034-4257(89)90091-6, 1989a.*

*Cho, E., Jacobs, J.M., Schroeder, R., Tuttle, S. E. and Olheiser, C.: Improvement of operational airborne gamma radiation using SMAP soil moisture, Remote Sens. Environ., 240, 111668, https://doi.org/10.1016/j.rse.2020.111668, 2020a.*

*Cho, E., Jacobs, J.M., and Vuyovich, C.: The value of long-term (40 years) airborne gamma radiation SWE record for evaluating three observation-based gridded SWE datasets by seasonal snow and land cover classifications, Water Resour. Res., 56, e2019WR025813, https://doi.org/10.1029/2019WR025813, 2020b.*

*Cork, H. F. and H. S. Loijens, H.S.: The effect of snow drifting on gamma survey results, J. Hydrol., 48(1-2), 41-51, https://doi.org/10.1016/0022-1694(80)90064-5, 1980.*

*Glynn, J.E., Carroll, T.R., Holman, P.B., Grasty, R.L.: An airborne gamma ray snow survey of a forested covered area with a deep snowpack, Remote Sens. Environ., 26 (2), 149 -160, https://doi.org/10.1016/0034-4257(88)90093-4, 1980.*

*Vogel, R. M., Carroll, T. R., and Carroll, S. S.: Simulation of airborne snow water equivalent measurement errors made over a forest environment, Proceedings of the American Society of Civil Engineers Symposium, Denver, CO, p. 9, 1985.*

3) Other snow-related data sources, such as UA-SWE data for 4-km resolution over CONUS and satellite snow cover data, could also be discussed in the paper.

As outlined above, the dataset's focus is in-situ SWE observations so UA-SWE and other CONUS-specific satellite snow cover data are beyond the scope of the dataset. Datasets such as NorSWE can help inform these spatially and temporally continuous datasets. We added text to Section 7 in this regard.

L443-446: *"In situ SWE observations can also help inform high quality spatially and temporally continuous gridded SWE estimates, exemplified by the University of Arizona SWE product (Broxton et al., 2019). SWE observations from SNOTEL and SD from the US Cooperative network are assimilated with gridded temperature and precipitation (PRISM) data to provide high-quality daily SWE and SD estimates over the Coterminous US at a 4 km spatial resolution (Zeng et al., 2018). Data contained in NorSWE could help facilitate the creation of similar datasets at both regional and hemispheric scales."*

Added references

Broxton, P., Zeng, X. and Dawson, N.: Daily 4 km Gridded SWE and Snow Depth from Assimilated In-Situ and Modeled Data over the Conterminous US. (NSIDC-0719, Version 1). [Data Set]. Boulder, Colorado USA. NASA National Snow and Ice Data Center Distributed Active Archive Center. https://doi.org/10.5067/0GGPB220EX6A, 2019.

Zeng, X., Broxton, P., Dawson, N.: Snowpack change from 1982 to 2016 over Conterminous United States, Geophys. Res. Lett., 45,12,940-12,947, https://doi.org/10.1029/2018GL079621, 2018.

**essd-2024-602 Author response to referee comment 3**

Original comments are in black, our responses are in blue, proposed additions and modifications in red. Line numbers refer to the original manuscript.

**Reviewer 3: Alexander Gottlieb**

This paper presents a novel dataset of in situ snow water equivalent (SWE) measurements from North America, Finland, and Russia over the period 1979-2021. The authors clean and harmonize data from 9 distinct sources to provide these measurements in a single file. This represents, to the best of my knowledge, the first publicly-available quasi-hemispheric in situ SWE dataset. As such, I believe it will be of tremendous value to the scientific community and the methods the authors apply seem to have produced a very high-quality dataset. Overall, I would not hesitate to recommend this paper for publication soon. I do, however, have a few minor suggestions that the authors could consider to strengthen the paper even further.

First, while I recognize the tremendous amount of work that goes into wrangling all of this SWE data, there are 2 additional sources that are fairly accessible that could increase the spatial coverage over Europe, particularly in mountainous regions. One is the GCOS dataset from Switzerland, available at https://envidat.ch/#/metadata/gcos-swe-data (citation below) The other is from the Norwegian Water Resources and Energy Directorate: https://www.nve.no/vann-og-vassdrag/vannets-kretsloep/snoe/automatiske-snostasjoner/?ref=mainmenu

Second, the authors do an excellent job of outlining possible uses for their dataset, and there is one more they can consider adding. For applications where spatiotemporally continuous data is necessary, the interpolation of in situ SWE and snow depth data has created some of the highest-quality estimates, such as the 4km University of Arizona SWE product (Broxton et al., 2019). The NorSWE dataset the authors present could facilitate the creation of a high-quality gridded hemispheric products that would be of tremendous value to the research community, and I would encourage them to highlight that as a potential use case.

Overall, I think this is a high-quality dataset that will be very useful, and look forward to the publication of the paper and the use of the data.

Broxton, P., Zeng, X. & Dawson, N. (2019). Daily 4 km Gridded SWE and Snow Depth from Assimilated In-Situ and Modeled Data over the Conterminous US. (NSIDC-0719, Version 1). [Data Set]. Boulder, Colorado USA. NASA National Snow and Ice Data Center Distributed Active Archive Center. https://doi.org/10.5067/0GGPB220EX6A.

Marty, C. (2020). GCOS SWE data from 11 stations in Switzerland. EnviDat. https://www.doi.org/10.16904/15.

Thank you for your constructive feedback and suggestions of additional data and use cases. As suggested also by Adrià Fontrodona-Bach (Reviewer 1) we added data from the Norwegian Water Resources and Energy Directorate (NVE) and the Swiss Federal Research Institute (WSL) – Institute for Snow and Avalanche Research (SLF): *'GCOS SWE data from 11 stations in Switzerland'* (Marty 2020). The NVE data includes six snow courses and 19 automated sites, one of which is located in

Nepal. All tables, figures and relevant text has been updated accordingly. The Norway (NVE) data rea detailed in Section 3.5 (L254-265). The Swiss data do not fit our criteria of snow courses or airborne measurements for SWE but are exceptionally included to address the data gap in Europe. The Single point manual measurements – Switzerland are detailed in Section 2.1 (L109-114).

*Marty, Christoph (2020). GCOS SWE data from 11 stations in Switzerland. EnviDat.doi:10.16904/15.*

You make an excellent point about high-resolution gridded datasets that we did not adequately highlight. Indeed, the University of Arizona SWE dataset is an excellent example of this. We propose to add this use case to Section 7.

[revised manuscript text omitted]